# Generalization Error Bounds for Learning under Censored Feedback

## Abstract

Generalization error bounds from learning theory provide statistical guarantees on how well an algorithm will perform on previously unseen data. In this paper, we characterize the impacts of data non-IIDness due to censored feedback (a.k.a. selective labeling bias) on such bounds. We first derive an extension of the well-known Dvoretzky-Kiefer-Wolfowitz (DKW) inequality, which characterizes the gap between empirical and theoretical CDFs given *IID* data, to problems with *non-IID data due to censored feedback*. We then use this CDF error bound to provide a bound on the generalization error guarantees of a classifier trained on such non-IID data. We show that existing generalization error bounds (which do not account for censored feedback) fail to correctly capture the model's generalization guarantees, verifying the need for our bounds. We further analyze the effectiveness of (pure and bounded) exploration techniques, proposed by recent literature as a way to alleviate censored feedback, on improving our error bounds. Together, our findings illustrate how a decision maker should account for the trade-off between strengthening the generalization guarantees of an algorithm and the costs incurred in data collection when future data availability is limited by censored feedback.

## 1 Introduction

Generalization error bounds are a fundamental concept in machine learning, which provide (statistical) guarantees on how a machine learning algorithm trained on some given dataset will perform on new, unseen data. However, many implicit or explicit assumptions about training data are often made when training ML models and deriving theoretical guarantees for their performance. These assumptions include access to independent and identically distributed (IID) training data, the availability of correct labels, and static underlying data distributions (Bartlett & Mendelson, 2002; Bousquet & Elisseeff, 2002; Cortes et al., 2019; 2020). Some studies in this area, e.g. Mohri & Rostamizadeh (2007; 2008); Kuznetsov & Mohri (2017); Cheng et al. (2018), have provided bounds when these assumptions are removed. In this paper, we are similarly interested in the impact of non-IID training data, specifically due to *censored feedback*, on the learned algorithm's generalization error guarantees.

Censored feedback, also known as selective labeling bias, arises in many applications wherein human or algorithmic decision-makers set certain thresholds or criteria for favorably classifying individuals, and subsequently only observe the true label of individuals who pass these requirements. For example, schools may require a minimum GPA or standardized exam score for admission; yet, graduation rates are only observed for admitted students. Financial institutions may set limits on the minimum credit score required for loan approval; yet, loan return rates are only observed for approved applicants. In these types of classification tasks, the algorithm's training dataset grows over time (as students are admitted, loans are granted); however, the new data is selected in a non-IID manner from the underlying domain, due to the unobservability of the true label of rejected data. This type of bias also arises when determining recidivism in courts, evaluating the effectiveness of medical treatments, flagging fraudulent online credit card transactions, etc. Despite this ubiquity, to the best of our knowledge, generalization error bounds given non-IID training data due to censored feedback remain unexplored. We close this gap by providing such bounds in this work, show the need for them, and formally establish the extent to which censored feedback hinders generalization.

One of the commonly proposed methods to alleviate the impacts of censored feedback is to *explore* the data domain, and admit (some of) the data points that would otherwise be rejected, with the goal of expanding the training data. Existing approaches to exploration can be categorized into *pure exploration* (Nie et al., 2018; Bechavod et al., 2019; Kazerouni et al., 2020; Kilbertus et al., 2020), where any individual in the exploration range may be admitted (with some probability $\epsilon$), and *bounded exploration* (Balcan et al., 2007; Wei, 2021; Yang et al., 2022; Lee et al., 2023), in which the exploration range is further limited based on cost or informativeness of the new samples. The additional data samples collected through (pure or bounded) exploration may not only help improve the accuracy of the learned model when evaluated on a given test data (as shown by these prior works), but may also help tighten the generalization error guarantees of the learned model; we formalize the latter improvement, and show how the frequency and range of exploration can be adjusted accordingly.

We note that censored feedback may or may not be avoidable depending on the application (given, e.g., the costs or legal implications of exploration). We therefore present generalization error bounds both with and without exploration, establishing the extent to which the decision maker should be concerned about censored feedback's impact on the learned model's guarantees, and how well they might be able to alleviate it if exploration is feasible.

**Our approach.** We characterize the generalization error bounds as a function of the gap between the empirically estimated cumulative distribution function (CDF) obtained from the training data, and the ground truth underlying distribution of data. At the core of our approach is noting that although censored feedback leads to training data being sampled in a non-IID fashion from the true underlying distribution, this non-IID data can be split into IID "subdomains". Existing error bounds for IID data, notably the Dvoretzky-Kiefer-Wolfowitz (DKW) inequality (Dvoretzky et al., 1956; Massart, 1990), can provide bounds on the deviation of the empirical and theoretical subdomain CDFs, as a function of the number of available data samples in each subdomain. The challenge, however, lies in reassembling such subdomain bounds into an error bound on the full domain CDFs. Specifically, this will require us to shift and/or scale the subdomain CDFs, with shifting and scaling factors that are themselves empirically estimated from the underlying data, and can be potentially re-estimated as more data is collected. Our analysis identifies these factors, and highlights the impacts of each on the error bounds.

**Summary of findings and contributions:**

1. We generalize the well-known Dvoretzky-Kiefer-Wolfowitz (DKW) inequality, which characterizes the gap between empirical and theoretical CDFs given *IID* data, to problems with *non-IID data due to censored feedback* without exploration (Theorem 2) and with exploration (Theorem 3), and formally show the extent to which censored feedback hinders generalization.

2. We characterize the change in these error bounds as a function of the severity of censored feedback (Proposition 1) and the exploration frequency (Proposition 2). We further show (Section 3.3) that a minimum level of exploration is needed to tighten the error bound.

3. We derive a generalization error bound (Theorem 4) for a classification model learned in the presence of censored feedback using the CDF error bounds in Theorems 2 and 3.

4. We numerically illustrate our findings (Section 5). We show that existing generalization error bounds (which do not account for censored feedback) fail to correctly capture the generalization error guarantees of the learned models. We also illustrate how a decision maker should account for the trade-off between strengthening the generalization guarantees of an algorithm and the costs incurred in data collection for reaching enhanced learning guarantees.

**Related works**. Although existing literature has studied generalization error bounds for learning from non-IID data, non-IIDness raised by censored feedback has been overlooked. Here, we discuss works most closely related to ours. We also provide a more detailed review of other related work in Appendix A.

First, our work is closely related to generalization theory in the PAC learning framework in non-IID settings, including (Mohri & Rostamizadeh, 2007; 2008; Yu, 1994) and (Kuznetsov & Mohri, 2017); these works consider dependent samples generated through a stationary, and non-stationary $\beta$-mixing sequence, respectively, where the dependence between samples weakens over time. To address the vanishing dependence issue, these works consider building blocks within which the data can be viewed as IID. The study of Yu (1994) is based

on the VC-dimension, while Mohri & Rostamizadeh (2008) and Mohri & Rostamizadeh (2007) focus on the Rademacher complexity and algorithm stability, respectively. Our work is similar in that we also consider identifying IID blocks within the data to circumvent data non-IIDness. However, we differ in our reassembly method, in the source of data non-IIDness, and in our study of the impacts of exploration. Furthermore, our bounds differ conceptually. While their bounds, based on the mixing parameter $\beta$, converge to zero by treating random samples across identified blocks as IID, our bounds are derived from threshold-based data collection. They are constructed by reassembling multiple IID blocks while explicitly accounting for the effects of censored feedback.

Our work is also closely related to partitioned active learning, including Cortes et al. (2019; 2020); Lee et al. (2023); Zheng et al. (2019). Cortes et al. (2019) partition the entire domain to find the best hypothesis for each subdomain, and a PAC-style generalization bound is derived compared to the best hypothesis over the entire domain. This idea is further extended to adaptive partitioning in Cortes et al. (2020). In Lee et al. (2023), the domain is partitioned into a fixed number of subdomains, and the most uncertain subdomain is explored to improve the mean-squared error. The work of Zheng et al. (2019) considers a special data non-IIDness where the data-generating process depends on the task property, partitions the domain according to the task types, and analyzes each subdomain separately. Our work is similar to these studies in that we also consider (active) exploration techniques, and partition the data domain to build IID blocks. However, we differ in problem setup and analysis approach, and in accounting for the cost of exploration when we consider bounded exploration techniques. More specifically, their bounds are derived from the aggregation of multiple subdomains with requested labels (we refer to as exploration). The key distinction lies in data availability: while they can request labels from any subdomain without considering the cost of doing so, we are constrained to explore samples from certain subdomains due to the presence of censored feedback.

The technique of identifying IID-blocks within non-IID datasets has also been used in other (application) contexts to address the challenge of generalization guarantees given non-IID data. For instance, Wang et al. (2023) investigate generalization performance with covariate shift and spatial autocorrelation in geostatistical learning. They address the non-IIDness issue by removing samples from the buffer zone to construct spatially independent folds. Similarly, Tang et al. (2021) study generalization performance within the Federated Learning paradigm with non-IID data. They employ clustering techniques to partition clients into distinct clusters based on statistical characteristics, thus treating samples from clients within each cluster as IID and analyzing each cluster separately. We similarly explore generalization performance with non-IID data samples and employ the technique of identifying IID subdomains/blocks. However, we differ in the reason for the occurrence of non-IIDness, the setup of the problem, and our analytical approaches.

Lastly, our work is related to the broader literature on multi-armed bandit learning (Bubeck et al., 2012; Lattimore & Szepesvári, 2020), which deals primarily with the exploration and exploitation dilemma. This same trade-off emerges the context of online/machine learning, where a decision maker can obtain additional data to improve the generalization performance of its models, while at the same time risking incurring costs due to this data collection. In the general bandit problem, the decision maker explores "arms" (the available actions) in various ways, such as randomly using the $\epsilon$-greedy algorithm, by some form of highest uncertainty as in UCB algorithm, or by importance sampling approaches as in EXP3, etc. The key difference in our approach is that we consider *bounded* exploration (motivated by works such as (Balcan et al., 2007; Lee et al., 2023; Wei, 2021; Yang et al., 2022)), where a bound is set to limit the "arms" that are considered for exploration (here, ranges of data samples that may be admitted). This is because the cost of wrong decisions increases as samples further away from the current decision threshold are admitted, making some arms too costly for exploration. Furthermore, in the bandit literature, regret analysis is conducted to analyze the model performance compared to the best actions in hindsight. In contrast, we analyze the model performance from a different angle: our goal is to improve the generalization error guarantees (upper bound on the difference between the model's performance on training data and unseen testing data) by utilizing the newly collected samples through exploration.

## 2 Problem Setting

We consider a supervised learning setup where a learner (equivalently, the learning algorithm) selects a classifier based on an initial training dataset and subsequently uses it to make binary decisions (e.g., accept/reject) for new data samples arriving sequentially. We use a bank granting loans as a running example.

**Data representation.** Each data sample is represented as a pair $(x, y)$, where $x \in \mathcal{X} \subseteq \mathbb{R}$ is the feature used for decision-making (e.g., credit score), and $y \in \mathcal{Y} = \{0, 1\}$ is the true label indicating qualification status, with $y = 1$ denoting that the sample is qualified to receive a favorable decision (e.g., the applicant will repay the loan if granted). We denote the corresponding random variables as $X$ and $Y$, and we use $F^y(x) = \mathbb{P}(X \leq x | Y = y)$ to denote the cumulative distribution function (CDF) of $X$ conditional on $Y = y$, and $p_y = \mathbb{P}(Y = y)$ to denote the label portions in the population.

**The learning algorithm.** The learner begins with an initial (historical) realized training dataset consisting of $n_y$ IID samples[1] $\{x_i^y\}_{i=1}^{n_y}$ for each label $y \in \{0, 1\}$. This initial training dataset can be likened to a financial institution with an existing loan repayment history dataset consisting of a set number of defaults and on-time payoffs. Based on the initial dataset, the learner selects a threshold-based binary classifier $f_\theta(x) : \mathcal{X} \to \{0, 1\}$ (i.e., $f_\theta(x) = \mathbb{1}(x \geq \theta)$) to decide whether to accept or reject (equivalently, assign labels 1 or 0) incoming loan applications, where $\theta$ denotes the decision threshold (e.g., $\theta$ could be the minimum credit score to be approved for a loan)

**Assumption 1.** *The data samples $x$ are one-dimensional, and the classifier is threshold-based, $f_\theta(x)$.[2] Extensions to high-dimensional samples are discussed in Appendix J.*

**Censored vs. disclosed regions.** The decision threshold $\theta$ divides the data domain into two regions: the upper, *disclosed* region, where the true label of future admitted samples will become known to the learner, and the lower, *censored* region, where true labels are no longer observed. As new samples arrive, due to this censored feedback, additional data is only collected from the disclosed region of the data domain (e.g., we only find out if an individual repays the loan if it is granted the loan in the first place). This is what causes the non-IIDness of the (expanded) dataset: after new samples arrive, the training dataset consists of $n_y$ historical IID samples from both censored and disclosed regions on each label $y$, and an additional $k_y$ samples collected afterward from each label $y$, but only from the disclosed region, making the entire $n_y + k_y$ samples a non-IID subset of the respective label $y$'s data domain.

**Remark 1.** *Note that we assume the learner starts with one fixed realization of $\{n_y\}_{y\in\{0,1\}}$ data points, and therefore the decision threshold $\theta$ and the number of initial samples in the censored region $\{m_y\}_{y\in\{0,1\}}$ are (non-random) realized values[3]. For instance, the realized training dataset and decision threshold can be likened to a financial institution with an existing loan repayment history dataset consisting of a set number of defaults and on-time payoffs, and its initial decision threshold selected for approving future applications.*

**Assumption 2.** *The non-IID nature of the data arises solely from censored feedback, which biases the data collection process by restricting observed labels to samples in the disclosed region.*

**Remark 2.** *We note that there are two possible ways to interpret the additional samples $\{k_y\}_{y\in\{0,1\}}$: a posteriori (i.e., outcomes after collecting exactly $k_y$ new samples in the disclosed region), or a priori (i.e., possible values once a total of $T$ new samples arrive, only some of which will fall in the disclosed region). The former is a reasonable assumption if a learner has already collected samples under censored feedback, or alternatively, is willing to wait to collect the exact required number of samples until it can achieve a desired error bound. The latter is from the viewpoint of a learner contemplating potential outcomes if it waits for a total of $T$ new samples to arrive. We will present our new error bound under both interpretations.*

Formally, let $F^y(x)$ denote the theoretical (ground truth) CDF for label $y$ samples. Let $\alpha^y := F^y(\theta)$ be the theoretical fraction of samples in the censored region, and $m_y$ be the random number of the initial $n_y$ training samples from label $y$ samples located in the censored region. It is worth noting that $\frac{m_y}{n_y}$ can provide

---

[1]We assume that any non-IIDness is introduced due to censored feedback impacting subsequent data collection. Extension to initially biased training data is also possible but at the expense of additional notation.

[2]The single-dimensional features and threshold classifier assumptions are not too restrictive: Corbett-Davies et al. (2017, Thm 3.2) and Raab & Liu (2021) have shown that threshold classifiers can be optimal if multi-dimensional features can be appropriately converted into a one-dimensional scalar (e.g., with a neural network).

[3]We consider a random variable $\{m_y\}_{y\in\{0,1\}}$ in Appendix K.

Table 1: Notation Summary

| Symbol | Explanation |
| --- | --- |
| $(x, y)$ | Paired (feature, label) information of samples |
| $\theta$ | Decision threshold |
| $LB$ | Exploration lower bound |
| $n$ $(n_y)$ | Number of initial samples (from label $y$) |
| $l, m$ | Number of initial samples that fall below the $LB, \theta$ |
| $\alpha, \beta$ | Theoretical fraction of samples that fall below the $\theta, LB$ |
| $k$ | Additional samples collected under no-exploration case |
| $k_e, k_d$ | Additional samples collected under exploration case, where $k_e$ and $k_d$ represent samples collected in the exploration and disclosed regions |
| $T$ | Total number of sequential arriving samples |
| $p_y$ | Label portions in the populations $\mathbb{P}(Y = y)$ |
| $F^y, F_n^y$ | Theoretical and empirical CDF $\mathbb{P}(X \leq x \mid Y = y)$ on the full data domain |
| $G^y, G_m^y$ | Theoretical and empirical CDF $\mathbb{P}(X \leq x \mid Y = y)$ on the censored region |
| $E^y, E_{m-l+k_1}^y$ | Theoretical and empirical CDF $\mathbb{P}(X \leq x \mid Y = y)$ on the exploration region |
| $K^y, K_{n-m+k}^y$ | Theoretical and empirical CDF $\mathbb{P}(X \leq x \mid Y = y)$ on the disclosed region |
| $R(\theta), R_{emp}(\theta)$ | Expected and empirical risk incurred by an algorithm with a decision threshold $\theta$. |

an empirical estimate of $\alpha^y$, but the two are in general not equal. After new samples are collected, the learner has access to $n_y + k_y$ total samples from label $y$ samples, which are not identically distributed: $m_y$ are in the censored region, and $n_y - m_y + k_y$ are in the disclosed region. Let $F_{n_y+k_y}^y(x)$ denote the empirical CDF of the feature distribution for label $y$ samples based on these $n_y + k_y$ training data points. Our first goal is to provide an error bound, similar to the DKW inequality, of the discrepancy between $F_{n_y+k_y}^y(x)$ and the ground truth CDF $F^y(x)$, for each label $y$. We will then use these to bound the generalization error guarantees of the learned model from the (non-IID) $\{n_y + k_y\}_{y \in \{0,1\}}$ data points.

We summarize our problem setting's dynamics, main notation below:

• **Stage I: Initial Data.** The learner starts with $n_y$ initial data points, $\{x_i^y\}_{i=1}^{n_y}$, from each label $y \in \{0, 1\}$, drawn IID from the corresponding true underlying distributions with CDF $F^y(x)$. Accordingly, the learner selects a fixed decision threshold $\theta$. Given $\theta$, the $n_y$ samples for label $y$ are be divided into $m_y$ samples below $\theta$ ($m_y = |\{i : x_i^y < \theta\}|$, referred to as the censored region) and $n_y - m_y$ samples above $\theta$ (referred to as the disclosed region).

• **Stage II: Arrival of New Samples.** At each time $t$, a new sample arrives. Its true label is $\hat{y} = y$ with probability $p_y$, and its feature $\hat{x}$ is drawn uniformly at random from the corresponding conditional distribution with CDF $F^{\hat{y}}(x)$. The sample's feature $\hat{x}$ is observed, and it is admitted if and only if $\hat{x} \geq \theta$. Due to censored feedback, $\hat{y}$ will only be observed if the sample is admitted. When a sample is admitted, its data is used to expand the corresponding dataset of $y = \hat{y}$ samples to $\{x_1^y, \ldots, x_{n_y}^y, x_{n_y+1}^y, \ldots, x_{n_y+k_y^t-1}^y, \hat{x}\}$.

• **Stage III: Updating Empirical Distribution Estimates.** After $T$ time steps (which can be fixed in advance, or denote the time at which a certain number of samples have been collected), the learner has access to $k_y$ new samples for each label $y$. These samples expand the training dataset for label $y$ into the non-IID collection $\{x_1^y, \ldots, x_{n_y}^y, x_{n_y+1}^y, \ldots, x_{n_y+k_y-1}^y, x_{n_y+k_y}^y\}$. The learner then find $F_{n_y+k_y}^y(x)$, the empirical CDF of the feature distribution for label $y$ based on the combined $n_y + k_y$ data points.

For a clearer understanding of the notations we used, we summarize all notations in the following Table 1.

## 3 Error Bounds on Cumulative Distribution Function Estimates

Recall that our first goal is to provide an error bound, similar to the DKW inequality, of the discrepancy between the empirical CDF of feature distribution $F_{n_y+k_y}^y(x)$ and the ground truth CDF $F^y(x)$, for each label $y$. Note that the empirical CDF is found for each label $y$ separately based on its own data samples.

Therefore, we drop the label $y$ from our notation throughout this section for simplicity. Further, we first derive the *a posteriori* bounds for given realizations of $k_y$, and develop the *a priori* version of the bound accordingly in Corollary 1 (c.f. Remark 2 for the distinction between *a priori* and *a posteriori* cases).

We first state the Dvoretzky-Kiefer-Wolfowitz inequality (an extension of the Vapnik–Chervonenkis (VC) inequality for real-valued data) which provides a CDF error bound given IID data.

**Theorem 1** (The Dvoretzky-Kiefer-Wolfowitz (DKW) inequality (Dvoretzky et al., 1956; Massart, 1990)). *Let $Z_1, \ldots, Z_n$ be IID real-valued random variables with cumulative distribution function $F(z) = \mathbb{P}(Z_1 \leq z)$. Let the empirical distribution function be $F_n(z) = \frac{1}{n} \sum_{i=1}^{n} \mathbb{1}(Z_i \leq z)$. Then, for every $n$ and $\eta > 0$,*

$$\mathbb{P}\left(\sup_{z \in \mathbb{R}} \left| F(z) - F_n(z) \right| \geq \eta\right) \leq 2 \exp\left(-2n\eta^2\right).$$

In words, the DKW inequality shows how the likelihood that the maximum discrepancy between the empirical and true CDFs exceeds a tolerance level $\eta$ decreases in the number of (IID) samples $n$.

**Bounds behavior**. To better understand the behavior of the bounds, we conduct a numerical experiment to illustrate why the bounds derived using DKW inequality fails to address the problem in the presence of censored feedback. We proceed as follows: we start with $n = 100$ initial random samples drawn from a Gaussian distribution with mean $\mu = 7$ and standard deviation $\sigma = 3$, with an additional $T = 10000$ samples arriving subsequently, randomly sampled from across the entire data domain. For significance level $\delta = 0.01$, we observe that the bounds decrease as more samples are collected over

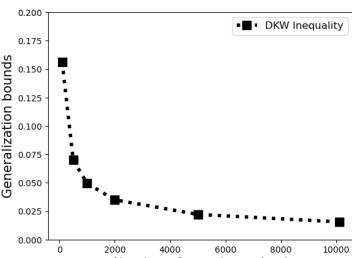

Figure 1: Behavior of bounds using DKW inequality.

the entire data domain. Furthermore, in accordance with the strong law of large numbers, the bounds eventually converge to zero as $T$ becomes sufficiently large.

However, when a decision threshold is set (e.g., $\theta = 8$)—indicating that only samples above the threshold are collected to improve the bounds—the situation changes. In this case, the collected samples are no longer representative of the entire data domain, as they are limited to the disclosed region. Consequently, the bounds associated with the censored region cannot be improved. This results in a persistent gap between the theoretical and empirical CDFs, preventing convergence to zero. In contrast, the bounds derived from the DKW inequality, which assume random sampling across the entire data domain, continue to decrease to zero. As a result, the bounds (represented by the black curve) eventually underestimate the true generalization error, crossing below it and failing to account for the censored feedback.

We now extend the DKW inequality to the case of non-IID data due to censored feedback. We do so by first splitting the data domain into blocks containing IID data, to which the DKW inequality is applicable. Specifically, although the expanded training dataset is non-IID, the decision maker has access to $m$ IID samples in the censored region, and $n - m + k$ IID samples in the disclosed region. Let $G_m$ and $K_{n-m+k}$ denote the corresponding empirical feature distribution CDFs. The DKW inequality can be applied to bound the difference between these empirical CDFs and the corresponding ground truth CDFs $G$ and $K$.

It remains to identify a connection between the full CDF $F$, and $G$ (the censored CDF) and $K$ (the disclosed CDF), to reach a DKW-type error bound on the full CDF estimate (see Figure 2 for an illustration). This reassembly from the bounds on the IID blocks into the full data domain is however more involved, as it requires us to consider a set of scaling and shifting factors, which are themselves empirically estimated and different from the ground truth values. We will account for these differences when deriving our generalization of the DKW inequality, as detailed in the remainder of this section. All proofs are given in the Appendix.

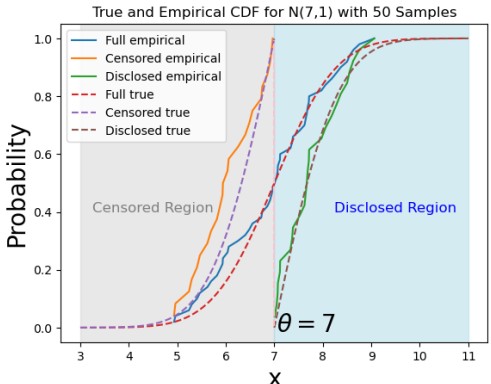

Figure 2: The empirical CDFs $F_{n+k}$ (Full domain), $G_m$ (Censored region), and $K_{n-m+k}$ (Disclosed region), and the theoretical CDFs of $F$, $G$, and $K$. Experiments based on randomly drawn samples from Gaussian data $N(7, 1)$, $\theta = 7$, $n = 50$, $m = 24$, and $k = 0$.

### 3.1 CDF bounds under censored feedback

We first present two lemmas that establish how the deviation of $G_m$ and $K_{n-m+k}$ from their corresponding theoretical values, for a given realization of $m$ data points in the censored region, relate to the deviation of the full empirical CDF $F_{n+k}$ from its theoretical value $F$.

**Lemma 1** (Censored Region). *Let $Z = \{x_i | x_i \leq \theta\}$ denote the (realized) $m$ out of $n + k$ samples that are in the censored region. Let $G$ and $G_m$ be the theoretical and empirical CDFs of $Z$, respectively. Then,*

$$\sup_{x \in (-\infty, \theta)} |F(x) - F_{n+k}(x)| \leq \sup_{x \in (-\infty, \theta)} \underbrace{\left| \min\left(\alpha, \frac{m}{n}\right)(G(x) - G_m(x)) \right|}_{\text{(scaled) censored subdomain error}} + \underbrace{\left| \alpha - \frac{m}{n} \right|}_{\text{scaling error}} .$$

The (partial) error bound in this lemma shows the maximum difference between the true $F$ and the empirical $F_{n+k}$ in the censored region (i.e., for $x \in (-\infty, \theta)$) can be bounded by the maximum difference between $G$ and $G_m$, modulated by the *scaling* ($\min(\alpha, \frac{m}{n})$) that is required to map from partial CDFs to full CDFs.

Specifically, to match the partial and full CDFs, we need to consider the different endpoints of the censored region's CDF and the full CDF at $\theta$, which are $G_m(\theta) = G(\theta) = 1$, $F(\theta) = \alpha$, and $F_{n+k}(\theta) = \frac{m}{n}$, respectively. The first term in the bound above accounts for this by scaling the deviation between the true and empirical partial CDF accordingly. The second term accounts for the error in this scaling since the empirical estimate $\frac{m}{n}$ is generally not equal to the true endpoint $\alpha$.

The following is a similar result in the disclosed region.

**Lemma 2** (Disclosed Region). *Let $Z = \{x_i | x_i \leq \theta\}$ denote the (realized) $n - m + k$ out of the $n + k$ samples in the disclosed region. Let $K$ and $K_{n-m+k}$ be the theoretical and empirical CDFs of $Z$, respectively. Then,*

$$\sup_{x \in (\theta, \infty)} |F(x) - F_{n+k}(x)| \leq \sup_{x \in (\theta, \infty)} \underbrace{\left| \min(1 - \alpha, 1 - \frac{m}{n})(K(x) - K_{n-m+k}(x)) \right|}_{\text{(scaled) disclosed subdomain error}} + \underbrace{2\left| \alpha - \frac{m}{n} \right|}_{\text{shifting and scaling errors}}$$

Similar to Lemma 1, we observe the need for a scaling factor. However, in contrast to Lemma 1, this lemma introduces an additional *shifting error*, resulting in a factor of two in the last term $|\alpha - \frac{m}{n}|$. In particular, we need to consider the different starting points of the disclosed region's CDF and full CDF at $\theta$, which are $K_m(\theta) = K(\theta) = 0$, $F(\theta) = \alpha$, and $F_{n+k}(\theta) = \frac{m}{n}$, respectively, when mapping between the CDFs; one of the $|\alpha - \frac{m}{n}|$ captures the error of shifting the starting point of the partial CDF to match that of the full CDF.

We can now state our main theorem, which generalizes the well-known DKW inequality to problems with censored feedback.

**Theorem 2.** *Let $x_1, x_2, \ldots, x_n$ be realized initial data samples, drawn IID from a distribution with CDF $F(x)$. Let $\theta$ partition the data domain into two regions, such that $\alpha = F(\theta)$, and $m$ of the initial $n$ samples are located to the left of $\theta$. Assume we have collected $k$ additional samples above the threshold $\theta$, and let $F_{n+k}(x)$ denote the empirical CDF estimated from these $n + k$ (non-IID) data. Then, for every $\eta > 0$,*

$$\mathbb{P}\left[\sup_{x \in \mathbb{R}}\left|F(x) - F_{n+k}(x)\right| \geq \eta\right] \leq \underbrace{2\exp\left(\frac{-2m(\eta - |\alpha - \frac{m}{n}|)^2}{\min\left(\alpha, \frac{m}{n}\right)^2}\right)}_{\text{censored region error (constant)}} + \underbrace{2\exp\left(\frac{-2(n-m+k)(\eta - 2|\alpha - \frac{m}{n}|)^2}{\min\left(1-\alpha, \frac{n-m}{n}\right)^2}\right)}_{\text{disclosed region error (decreasing with additional data)}}$$

The proof proceeds by applying the DKW inequality to each subdomain, and combining the results using a union bound on the results of Lemmas 1 and 2.

From the above expression, we observe that as $n$ (the number of initial samples across the entire data domain) becomes large, the maximum discrepancy between the theoretical and empirical CDFs decreases, following the strong law of large numbers. In such cases, the effect of censored feedback on the bounds becomes minimal, as the data distribution can already be well estimated. More interestingly, when the initial training dataset is small, censored feedback can have a substantial impact on the bounds as shown in Fig. 3(a). As the number of samples collected under censored feedback increases ($k \to \infty$), the error term associated with the censored region remains constant, as it does not depend on $k$. However, the error term for the disclosed region decreases asymptotically, behaving as $2exp(-2k\eta^2)$, as shown in Fig. 3(b). Together, this results in an overall decreasing trend, which is also reflected in the numerical illustration in Fig. 5, where the orange line falls below the blue line. However, unlike the DKW bound, this error bound does not go to zero due to a constant error term from the censored region of the data domain (the first term in the error bound). This means that unless exploration strategies are adopted, we can not guarantee arbitrarily good generalization in censored feedback tasks. Finally, we note that the DKW inequality can be recovered as the special case of our Theorem 2 by letting $\theta \to -\infty$ (which makes $\alpha \approx 0, m \approx 0$).

**Numerical Illustration of Bound Behavior for Each Term in Theorem 2.** We conduct a numerical experiment to illustrate the behavior of the bounds for each term. We proceed as follows: we start with $n \in \{50, 200, 500, 1000\}$ initial random samples drawn from a Gaussian distribution with mean $\mu = 7$ and standard deviation $\sigma = 3$. For significance levels $\delta \in \{0.01, 0.05, 0.1\}$, Fig. 3(a) shows that, for any fixed initial training sample size $n$, the CDF bounds for the censored region remain constant. Since this term does not depend on newly collected samples, the bounds can only be improved by increasing $n$. For the disclosed region, we fix $n = 50$ and examine how the bounds behave as the number of newly collected samples varies ($k \in \{0, 50, 200, 500\}$). As shown in Fig. 3(b), for significance levels $\delta \in \{0.01, 0.05, 0.1\}$, the CDF bounds for the disclosed region vanish as more samples are collected.

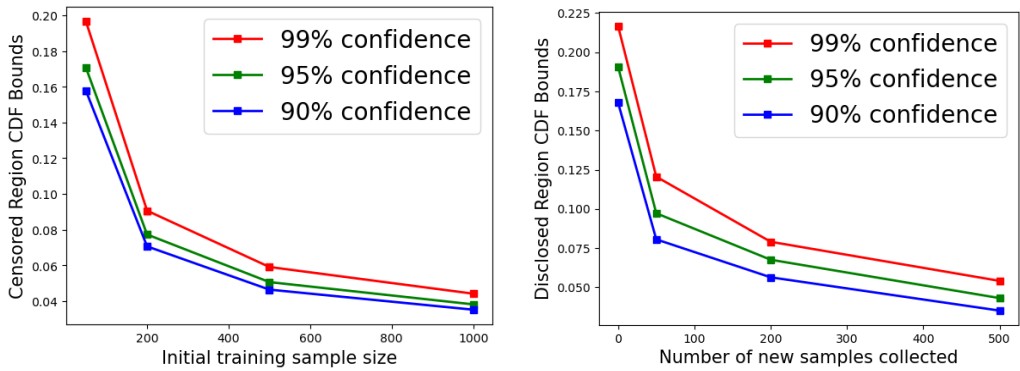

Figure 3: Behavior of the censored and disclosed region error terms.

Finally, recall from Remark 2 that instead of considering an exact realization of $k$ new samples in the disclosed region, a decision maker may want to know the error bound after waiting for $T$ agents to arrive

(only some of which will fall in the disclosed region). The following corollary provides an error bound that can be leveraged under this viewpoint.

**Corollary 1.** *Let $x_1, x_2, \ldots, x_n$ be realized initial data samples, drawn IID from a distribution with CDF $F(x)$. Let $\theta$ partition the data domain into two regions, such that $\alpha = F(\theta)$, and $m$ of the initial $n$ samples are located to the left of $\theta$. Assume we have waited for $T$ additional samples to arrive, and let $F_{n+T}(x)$ denote the empirical CDF estimated accordingly. Then, for every $\eta > 0$,*

$$\mathbb{P}\left[\sup_{x \in \mathbb{R}}\left|F(x) - F_{n+T}(x)\right| \geq \eta\right] \leq \underbrace{2\exp\left(\frac{-2m(\eta - |\alpha - \frac{m}{n}|)^2}{\min\left(\alpha, \frac{m}{n}\right)^2}\right)}_{\text{censored region error (constant)}}$$

$$+ \underbrace{\sum_{k=0}^{T} 2\binom{T}{k}(1-\alpha)^k \alpha^{T-k}\exp\left(\frac{-2(n-m+k)(\eta - 2|\alpha - \frac{m}{n}|)^2}{\min\left(1-\alpha, \frac{n-m}{n}\right)^2}\right)}_{\text{disclosed region error (decreasing with wait time T)}}$$

The proof is straightforward, and follows from writing the law of total probability for the left-hand side of the inequality by conditioning on the realization $k$ of the samples in the disclosed region. We first note that the censored region error term, as expected, is unaffected by the wait time $T$. The second term is the disclosed region error from Theorem 2; it is decreasing with $T$ as the exponential error terms decrease with $k$, and higher $k$'s are more likely at higher $T$.

### 3.2 Censored feedback and exploration

A commonly proposed method to alleviate censored feedback, as noted in Section 1, is to introduce exploration in the data domain. From the perspective of the generalization error bound, exploration has the advantage of reducing the constant error term in Theorem 2, by collecting more data samples from the censored region. Formally, we consider (bounded) exploration in the *range* $x \in (LB, \theta)$, where samples in this range are admitted with an exploration *frequency* $\epsilon$. When $LB \to -\infty$, this is a pure exploration strategy.

Now, the lowerbound $LB$ and the decision threshold $\theta$ partition the data domain into three IID subdomains (see Figure 4 for an illustration). However, the introduction of the additional *exploration region* $(LB, \theta)$ will enlarge the CDF bounds, as it introduces new scaling and shifting errors when reassembling subdomain bounds into full domain bounds.

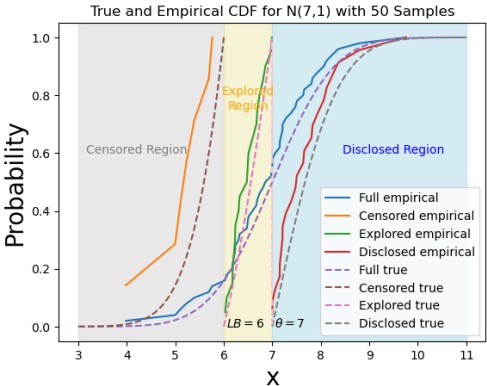

Figure 4: The empirical CDFs $F_{n+k_e+k_d}$ (Full domain), $G_l$ (Censored region), $E_{m-l+k_e}$ (Explored region), and $K_{n-m+k_d}$ (Disclosed region), and the theoretical CDFs of $F, G, E$, and $K$. Experiments based on randomly drawn samples from Gaussian data $N(7, 1)$, $\theta = 7$ $LB = 6$, $n = 50$, $l = 7$, $m = 27$, and $k_e = k_d = 0$.

Specifically, of the $n$ initial data, let $l$, $m - l$, and $n - m$ of them be in the censored (below $LB$), exploration (between $LB$ and $\theta$), and disclosed (above $\theta$) regions, respectively. Let $\beta = F(LB)$ and $\alpha = F(\theta)$, with initial empirical estimates $\frac{l}{n}$ and $\frac{m}{n}$, respectively. We will view both $l$ and $m$ as random variables.

As new agents arrive, let $k_e$ and $k_d$ denote the additional samples collected in the exploration range and disclosed range, respectively. One main difference of this setting with that of Section 3.1 is that as additional samples are collected, the empirical estimate of $\alpha$ can be re-estimated. Accordingly, we present a lemma similar to Lemmas 1 and 2 for the exploration region.

**Lemma 3** (Exploration Region). *Let $Z = \{x_i | LB \leq x_i \leq \theta\}$ denote the (realized) $m - l + k_e$ samples out of the $n + k_e + k_d$ samples that are in the exploration range. Let $E$ and $E_{m-l+k_e}$ be the theoretical and empirical CDFs of $Z$, respectively. Then,*

$$\sup_{x \in (LB, \theta)} |F(x) - F_{n+k_e+k_d}(x)| \leq \underbrace{\left| \beta - \frac{l}{n} \right|}_{\text{shifting error}} + \underbrace{\left| \alpha - \beta - \frac{n-l}{n} \frac{m-l+k_e}{n-l+k_e+\epsilon k_d} \right|}_{\text{re-estimated scaling error}}$$

$$+ \underbrace{\sup_{x \in (LB, \theta)} \left| \min\left( \alpha - \beta, \tfrac{n-l}{n} \tfrac{m-l+k_e}{n-l+k_e+\epsilon k_d} \right)(E(x) - E_{m-l+k_e}(x)) \right|}_{\text{scaled exploration subdomain error}}$$

Observe that here, we need both scaling and shifting factors to relate the partial and full CDF bounds, as in Lemma 2, but with an evolving scaling error as more data is collected. In particular, the initial empirical estimate $\frac{m}{n}$ is updated to $\frac{l}{n} + \frac{n-l}{n} \frac{m-l+k_e}{n-l+k_e+\epsilon k_d}$ after the observation of the additional $k_e$ and $k_d$ samples.

We now extend the DKW inequality when data is collected under censored feedback *and* with exploration.

**Theorem 3.** *Let $x_1, x_2, \ldots, x_n$ be realized initial data samples, drawn IID from a distribution with CDF $F(x)$. Let $LB$ and $\theta$ partition the domain into three regions, such that $\beta = F(LB)$ and $\alpha = F(\theta)$, with $l$ and $m$ of the initial $n$ samples located to the left of $LB$ and $\theta$, respectively. Assume we have collected an additional $k_e$ samples between $LB$ and $\theta$, under an exploration probability $\epsilon$, and an additional number of $k_d$ samples above $\theta$. Let $F_{n+k_e+k_2}(x)$ denote the empirical CDF estimated from these $n + k_e + k_d$ non-IID samples. Then, for every $\eta > 0$,*

$$\mathbb{P}\left[ \sup_{x \in \mathbb{R}} \left| F(x) - F_{n+k_e+k_d}(x) \right| \geq \eta \right] \leq \underbrace{2 \exp\left( \frac{-2l(\eta - |\beta - \frac{l}{n}|)^2}{\min\left( \beta, \frac{l}{n} \right)^2} \right)}_{\text{(still) censored region error (constant)}}$$

$$+ \underbrace{2 \exp\left( \frac{-2(m-l+k_e)\left( \eta - |\beta - \frac{l}{n}| - |\alpha - \beta - \frac{n-l}{n} \frac{m-l+k_e}{n-l+k_e+\epsilon k_d}| \right)^2}{\min\left( \alpha - \beta, \frac{n-l}{n} \frac{m-l+k_e}{n-l+k_e+\epsilon k_d} \right)^2} \right)}_{\text{exploration region error (decrease with } k_e)} + \underbrace{2 \exp\left( \frac{-2(n-m+k_d)\left( \eta - 2|\alpha - \frac{l}{n} - \frac{n-l}{n} \frac{m-l+k_e}{n-l+k_e+\epsilon k_d}| \right)^2}{\min\left( 1 - \alpha, \frac{n-l}{n} \frac{n-m+\epsilon k_d}{n-l+k_e+\epsilon k_d} \right)^2} \right)}_{\text{disclosed region error (decrease with } k_d)}.$$

Comparing this expression with Theorem 2, we first note that the last term corresponding to the error bound in the disclosed region are similar when setting $k = k_d$, with the difference being in the impact of re-estimating $\alpha$.

Theorem 3 provides an extension of the DKW inequality to account for censored feedback and exploration, introducing three distinct regions: the (still) censored region $(-\infty, LB)$, the exploration region $(LB, \theta)$, and the disclosed region $(\theta, \infty)$. The (still) censored region contributes a constant error term dependent on $l$, the number of initial samples in this region, due to the absence of exploration. The exploration region introduces $k_e$, the number of samples collected under an exploration probability $\epsilon$, which reduces the error in this region as $k_e \to \infty$, ultimately approaching zero. In contrast, the disclosed region contributes an error based on $n - m$, the number of initial samples above $\theta$, and $k_d$, the number of additional samples collected in this region. As $k_d$ increases, the error in the disclosed region also diminishes.

A key insight from Theorem 3 is that although there can still be a non-vanishing error term in the (still) censored region, additional samples collected in the exploration and disclosed regions can reduce their respective error terms. Similar to Theorem 2, due to the newly collected samples $k_e$ and $k_d$, the error term for the exploration and disclosed region decreases asymptotically, behaving as $2exp(-2k_e\eta^2)$ and $2exp(-2k_d\eta^2)$, respectively, similar to the findings in Fig. 3. However, as also noted in Fig. 5, the union bounds can be problematic when $k_e, k_d$ is small, where the red line initially is above the orange line with a small exploration

probability. Further, if we adopt pure exploration ($LB \to -\infty$, which makes $\beta \approx 0, l \approx 0$), the first term will vanish as well (however, note that pure exploration may not be a feasible option if exploration is highly costly). Lastly, we note that an *a priori* version of this bound can be derived using similar techniques to that of Corollary 1.

### 3.3 When will exploration improve generalization guarantees?

It might seem at first sight that the new vanishing error term in the exploration range of Theorem 3 necessarily translates into a tighter error bound than that of Theorem 2 when exploration is introduced. Nonetheless, the shifting and scaling factors, as well as the introduction of an additional union bound, enlarge the CDF error bound. Therefore, in this section, we elaborate on the trade-off between these factors, and evaluate when the benefits of exploration outweigh its drawbacks in providing error bounds on the data CDF estimates.

We begin by presenting two propositions that assess the change in the bounds of Theorems 2 and 3 as a function of the severity of censored feedback (as measured by $\theta$) and the exploration frequency $\epsilon$.

**Proposition 1.** *Let $B(\theta)$ denote the error bound in Theorem 2, and assume the conditions of that theorem hold. Assume also that we can collect an additional $k = O(n)$ samples above the threshold. Then, $B(\theta)$ is increasing in $\theta$.*

**Proposition 2.** *Let $B^e(LB, \theta, \epsilon)$ denote the error bound in Theorem 3, and assume the conditions of that theorem hold. Then, $B^e(LB, \theta, \epsilon)$ is decreasing in $\epsilon$.*

In words, as intuitively expected, these propositions state that the generalization bounds worsen (i.e., are less tight) when the censored feedback region is larger, and that they can be improved (i.e., made more tight) as the frequency of exploration increases.

**Numerical illustration**. We also conduct a numerical experiment to illustrate the bounds derived in Theorems 2 and 3. We proceed as follows: $n = 8000$ random samples are drawn from a Gaussian distribution with mean $\mu = 7$ and standard deviation $\sigma = 3$, with an additional $T = 40000$ samples arriving subsequently, randomly sampled from across the entire data domain. We set $\delta = 0.01$, the threshold $\theta = 8$, and the lower bound $LB = 6$. We run the experiment 5 times and report the error bounds from Theorems 2 and 3 accordingly.[4]

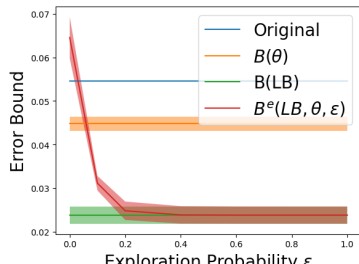

Figure 5: A minimum exploration frequency is needed to tighten the CDF error bound.

In Figure 5, the "original" (blue) line represents the DKW CDF bound of the initial samples without additional data. The "$B(\theta)$" (orange) line and "$B(LB)$" (green) line represent the CDF bound in Theorem 2 without exploration, where the decision threshold is at $\theta$ and $LB$, respectively. The "$B^e(LB, \theta, \epsilon)$" (red) line represents the bound in Theorem 3 with exploration probability $\epsilon$.

From Figure 5, we first observe that the green line ($B(LB)$, which observes new samples with $x \geq LB = 6$) provides a tighter bound than the orange line ($B(\theta)$, which observes new samples with $x \geq \theta = 8$), with both providing tighter bounds than the blue line (original DKW bound, before any new samples are observed). This is shown by that the green line is below the orange line, which is also below the blue line. This improvement is due to collecting more samples from the disclosed region results in a decrease in the CDF error bound, as noted by Proposition 1. Additionally, we can observe from the trajectory of the red line ($B^e(LB, \theta, \epsilon)$, which observes a fraction $\epsilon$ of new samples from $(LB, \theta)$, and all new samples above $\theta$) that introducing exploration enlarges the CDF error bound due to the additional union bound, but it also enables the collection of more samples, leading to a decrease in the CDF error bound as $\epsilon$ increases (evidenced by the red line is decreasing when $\epsilon$ increases along the x-axis); note that this observation aligns with Proposition 2.

---

[4]While we evaluate the *a posteriori* bounds under the realizations of the new samples falling in the disclosed and exploration regions ($k$, , $k_1$, and $k_2$), we show the averages of the bounds over multiple runs, which can be viewed as an approximation of the *a priori* version of the bounds. See Appendix L for a comparison of the bounds from Corollary 1 and Theorem 2.

Notably, we see that a minimum level of exploration probability $\epsilon$ (accepting around 10% of the samples in the exploration range) is needed to improve the CDF bounds over no exploration. Note that this may or may not be feasible for a decision maker depending on the costs of exploration (see also Section 3.4). However, if exploration is feasible, we also see that accepting around 20% of the samples in the exploration range (when the red line is close to the green line) can be sufficient to provide bounds nearly as tight as observing all samples in the exploration range. In other words, we can see from Fig. 5 that the $\epsilon$ is around 10% when the red line crosses the orange line, and it is around 20% when the red line is close to the green line.

### 3.4 How to choose an exploration strategy?

We close this section by discussing potential considerations in the choice of an exploration strategy in light of our findings. Specifically, a decision maker can account for a tradeoff between *the costs of exploration* and *the improvement in the generalization error bound* when choosing its exploration strategy. Recall that the exploration strategy consists of selecting an exploration lowerbound/range $LB$ and an exploration probability $\epsilon$. Formally, the decision maker can solve the following optimization problem to choose these parameters:

$$\max_{\epsilon \in [0,1], LB \in [0,\theta]} \left( B(\theta) - B^e(LB, \theta, \epsilon) \right) - C(LB, \theta, \epsilon) , \tag{1}$$

where $B(\theta)$ and $B^e(LB, \theta, \epsilon)$ denote the error bounds in Theorems 2 and 3, respectively, and $C(LB, \theta, \epsilon)$ is an exploration cost which is non-increasing in $(\theta - LB)$ (reducing the exploration range will weakly decrease the costs) and non-decreasing in $\epsilon$ (exploring more samples will weakly increase the cost). As an example, the cost function $C(LB, \theta, \epsilon)$ can be given by

$$C(LB, \theta, \epsilon) = \epsilon \int_{LB}^{\theta} e^{\frac{\theta - x}{c}} f^0(x) \mathrm{d}x. \tag{2}$$

In words, unqualified (costly) samples at $x$ have a density $f^0(x)$, and when selected (as captured by the $\epsilon$ multiplier), they incur a cost $e^{\frac{\theta - x}{c}}$, where $c > 0$ is a constant. Notably, observe that the cost is increasing as the sample $x$ gets further away from the threshold $\theta$. For instance, in the bank loan example, this could capture the assumption that individuals with lower credit scores default on a larger portion of their loans.

As noted in Proposition 2, $B^e(LB, \theta, \epsilon)$ is decreasing in $\epsilon$; coupled with any cost function $C(LB, \theta, \epsilon)$ that is (weakly) increasing in $\epsilon$, this means that the decision maker's objective function in equation 1 captures a tradeoff between reducing generalization errors and modulating exploration costs.

The optimization problem in equation 1 can be solved (numerically) by plugging in for the error bounds from Theorems 2 and 3 and an appropriate cost function (e.g., equation 2). For instance, in the case of the numerical example of Fig. 5, under the cost function of equation 2 with $c = 5$, and fixing $LB = 6$, the decision maker should select $\epsilon = 11.75\%$.

Another potential solution for modulating exploration costs is to use multiple exploration subdomains, each characterized by an exploration range $[LB_i, LB_{i-1})$, and with a higher exploration probability $\epsilon_i$ assigned to the subdomains closer to the decision boundary (which are less likely to contain high cost samples). For instance, with the choice of $b$ subdomains, the cost function of equation 2 would change to (the lower) cost:

$$C(\{LB_i\}_{i=1}^{b}, \theta, \{\epsilon_i\}_{i=1}^{b}) = \sum_{i=1}^{b} \epsilon_i \int_{LB_i}^{LB_{i-1}} e^{\frac{\theta - x}{c}} f^0(x) \mathrm{d}x. \tag{3}$$

It is worth noting that while this approach can reduce the costs of exploration, it will also weaken generalization guarantees when we reassemble the $b$ exploration subdomains' bounds back into an error bound of the full domain (similar to what was observed in Fig. 5 for $b = 1$). This again highlights a tradeoff between improving learning error bounds and restricting the costs of data collection.

## 4 Generalization Error Bounds under Censored Feedback

In this section, we use the CDF error bounds from Section 3 to characterize the generalization error of a classification model that has been learned from data collected under censored feedback. Specifically, we will

first establish a connection between the generalization error of a classifier (the quality of its learning) and the CDF error bounds on its training dataset (the quality of its data). With this relation in hand, we can then use any of the CDF error bounds from Theorems 1-3 to bound how well algorithms learned on data suffering from censored feedback (and without or with exploration) can generalize to future unseen data.

Formally, we consider a 0-1 learning loss function $\mathcal{L} : \mathcal{Y} \times \mathcal{Y} \to \{0, 1\}$. Denote $R(\theta) = \mathbb{E}_{XY} \mathcal{L}(f_\theta(X), Y)$ as the expected risk incurred by an algorithm with a decision threshold $\theta$. Similarly, we define the empirical risk as $R_{emp}(\theta)$. The *generalization error bound* is an upper bound to the error $|R(\hat{\theta}) - R_{emp}(\hat{\theta})|$, where $\hat{\theta}$ is the minimizer of the empirical loss, i.e., $\hat{\theta} := \arg\min_\theta R_{emp}(\theta)$. In words, the bound provides a (statistical) guarantee on the performance $R(\hat{\theta})$, when using the learned $\hat{\theta}$ on unseen data, relative to the performance $R_{emp}(\hat{\theta})$ assessed on the training data. Our objective is to characterize this bound under censored feedback, and to evaluate how utilizing (pure or bounded) exploration can improve the bound.

Recall that the decision maker starts with a training data containing $n_y$ IID samples from each label $y$, drawn from an underlying distribution with CDF $F^y(x)$. Let $n = n_0 + n_1$ denote the size of the initial training data. Then, the expected loss of a binary classifier with decision threshold $\theta$ is given by,

$$R(\theta) = \mathbb{E}_{XY} \mathcal{L}(f(X), Y) = p_1 F^1(\theta) + p_0(1 - F^0(\theta)) \ ,$$

while the empirical loss $R_{emp}(\theta)$ is given by,

$$R_{emp}(\theta) = \frac{n_1}{n} \frac{1}{n_1} \sum_{(x_i, y_i)} \mathbb{1}\{x_i \leq \theta, y_i = 1\} + \frac{n_0}{n} \left(1 - \frac{1}{n_0} \sum_{(x_i, y_i)} \mathbb{1}\{x_i \leq \theta, y_i = 0\}\right).$$

Similarly, if the decision maker can collect an additional $k_y$ samples of agents with features above the threshold $\theta$, the above empirical risk expression can be updated accordingly, by considering the $n_y + k_y$ samples available from each label $y$.

We detail the derivations of these expressions in Appendix I. Using these expressions of the expected and empirical risks, the following theorem provides an upper bound on the generalization error $|R(\hat{\theta}) - R_{emp}(\hat{\theta})|$ as a function of the CDF error bound, where $\hat{\theta}$ denotes the minimizer of the empirical loss, i.e., $\hat{\theta} := \arg\min_\theta R_{emp}(\theta)$.

**Theorem 4.** *Consider a threshold-based classifier $f_{\hat{\theta}}(x) : \mathcal{X} \to \{0, 1\}$ under a 0-1 loss function. Suppose we start with $n_y$ initial IID training samples from each label $y$, with $n = n_0 + n_1$. Let $\theta$ be a fixed/realized initial decision threshold calculated through one realization of the initial training dataset such that $f_\theta(x) = \mathbb{1}(x \geq \theta)$. Let $p_y$ denote the proportion of agents from label $y$. Subsequently, due to the censored feedback, the algorithm collects $k_y$ additional samples from each label $y$. Let $F^y$ and $F^y_m$ denote the CDFs and empirical CDFs, respectively, given $m$ samples from label $y$ agents. Then, with probability at least $1 - 2\delta$,*

$$\left| R(\hat{\theta}) - R_{emp}(\hat{\theta}) \right| \leq 3 \left| p_0 - \frac{n_0}{n} \right| + \sum_{y \in \{0,1\}} \min\left(p_y, \frac{n_y}{n}\right) \sup_\theta \left| F^y(\theta) - F^y_{n_y + k_y}(\theta) \right| \ .$$

The proof is given in Appendix H. First, we note that tightening the CDF error bounds leads to tightening the generalization error guarantees. More specifically, using this theorem together with Theorems 1, 2, and 3, we can provide a generalization error guarantee for an algorithm in terms of the number of available data samples in its training data from each label and in different parts of the data domain, particularly when future data availability is non-IID due to censored feedback.

For instance, the DKW inequality can be alternatively expressed as follows: given $n_y$ IID samples from a label $y$, with probability at least $1 - \delta$, the following inequality holds:

$$\sup_z \left| F(z) - F^y_{n_y}(z) \right| \leq \sqrt{\frac{\log \frac{2}{\delta}}{2n_y}} \ .$$

Using this expression in Theorem 4, we conclude that (without censored feedback, or with pure exploration with $\epsilon = 1$) with probability at least $1 - 2\delta$,

$$\left| R(\hat{\theta}) - R_{emp}(\hat{\theta}) \right| \leq 3\left| p_0 - \frac{n_0}{n} \right| + \sum_{y \in \{0,1\}} \min\left( p_y, \frac{n_y}{n} \right) \sqrt{\frac{\log \frac{2}{\delta}}{2n_y}}.$$

We can similarly specialize Theorem 4 to tasks with censored feedback by linking it with Theorems 2 and 3. Given the complexity of the CDF error bounds under censored feedback, while we cannot derive a closed-form expression for the bound as done for the DKW inequality, we can compute the bounds numerically, as shown in the next section.

## 5 Numerical Experiments

### 5.1 CDF error bounds

We first illustrate our derived bounds (with $\delta = 0.015$) on the empirical CDF. We start with 50 random samples from a Gaussian distribution N(7,1). Next, 200 new samples are drawn from the same distribution, with all samples with features $x \geq \theta = 7$ accepted, and samples with features $LB = 6 \leq x \leq \theta$ accepted with a probability $\epsilon \in \{0, 0.5, 1\}$; higher values of $\epsilon$ represent less censored feedback ($\epsilon = 1$ means no censored feedback).

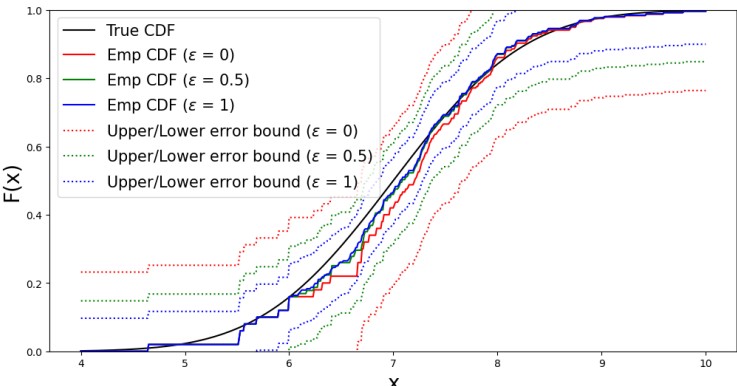

Figure 6: CDF error bounds when different levels of exploration ($\epsilon$) are used to alleviate censored feedback. As $\epsilon$ increases: (a) the empirical CDF estimates become more accurate, and (b) our CDF error bounds improve (i.e., more tightly enclose the true CDF).

From Figure 6, we first note that our bounds (the dotted lines) effectively enclose the true distribution, evidenced by that the true distribution is upper and lower bounded by the dotted lines. We also note the distinction between empirical CDFs in the disclosed region ($x \geq 7$) and the censored region ($x \leq 7$): as intuitively expected, empirical CDFs (solid lines) in the disclosed region are "smoother" compared to those in the censored region. Furthermore, as $\epsilon$ (exploration) increases, we overcome censored feedback in the exploration region, resulting in more accurate empirical estimates. Additionally, as $\epsilon$ increases, our error bounds improve (i.e., more tightly enclose the true CDF). In other words, we can see from Figure 6 that the empirical CDF in both explored and disclosed regions is getting smoother and closer to the true distribution with more samples collected. In addition, with a higher $\epsilon$, the gap between the upper and lower bounds is smaller and can still enclose the true distribution.

## 5.2 Model generalization error bounds: real-world data and adaptively updated algorithm

We now illustrate the ability of our generalization error bounds (derived in Theorem 4) in providing guarantees on the error of the learned models from data affected by censored feedback, using experiments on a real-world dataset: *FICO*(Hardt et al., 2016), *Retiring Adult*(Ding et al., 2021), *Adult*(Dua & Graff, 2017).

**Experiments on *FICO* dataset.** The *FICO* dataset is used to predict whether an individual will default. It includes one-dimensional features (e.g., credit scores) with a specific focus on the distribution information of the credit scores. We employ a logistic regression algorithm and 0-1 loss for the classification task, and compare the generalization error across different exploration probabilities ($\epsilon = \{0.5, 1\}$). We start with a 1000 training data samples. A total of 175000 new samples arrive throughout the experiment; in addition to accepting all samples with feature $x \geq \hat{\theta}$, the algorithm also accepts some samples that fall below $\hat{\theta}$. The decision threshold is updated periodically based on new data (after each 20000 batch of new samples arrives). The decision threshold is retrained using (most recent) training data. We report our experiment results for an average of 5 runs, where the randomness comes from the order of samples arrived and the exploration.

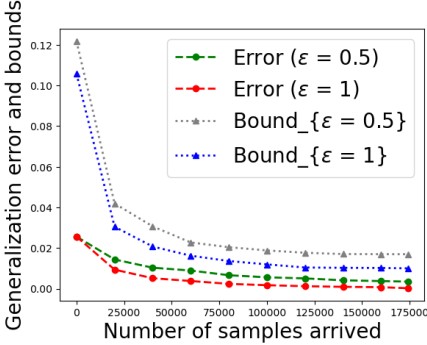

Figure 7: Generalization error and bounds using *FICO* dataset.

| | | Number of Samples Arrived | | | | | | |
|---|---|---|---|---|---|---|---|---|
| | | 0 | 20k | 40k | 80k | 120k | 160k | 175k |
| $\epsilon = 1$ | $|R(\hat{\theta}) - R_{emp}(\hat{\theta})|$ | 0.0257 | 0.0094 | 0.0052 | 0.0024 | 0.0013 | 0.0008 | 0.0003 |
| | Generalization Error Bounds | 0.106 | 0.031 | 0.021 | 0.014 | 0.010 | 0.010 | 0.010 |
| $\epsilon = 0.5$ | $|R(\hat{\theta}) - R_{emp}(\hat{\theta})|$ | 0.0257 | 0.0144 | 0.0104 | 0.0067 | 0.0051 | 0.0038 | 0.0035 |
| | Generalization Error Bounds | 0.122 | 0.042 | 0.031 | 0.021 | 0.0177 | 0.0170 | 0.0170 |

Table 2: Numerical summary table for the generalization error and bounds across the number of arrived samples using *FICO* dataset.

**Experiments on *Retiring Adult* dataset.** The *Retiring Adult* census dataset is used to predict whether an individual can earn more than $50k/year, based on a multi-dimensional feature set. Similar to the experiments on *FICO* dataset, we employ a logistic regression algorithm and 0-1 loss for the classification task, and compare the generalization error across different exploration probabilities ($\epsilon = \{0.5, 1\}$). A total of 1600000 new samples arrive throughout the experiment and the decision threshold is updated periodically based on new data (after each 200000 batch of new samples arrives). We report our experiment results for an average of 5 runs, where the randomness comes from the order of samples arrived and the exploration.

In Fig. 7 and 8, the y-axis represents the generalization error and its bounds, where a lower value is preferable. A smaller value of the bounds indicates tighter bounds enclosing the generalization error curve. From the experiment results from both *FICO* and *Retiring Adult* datasets in Fig. 7 and 8, we can see that our bounds (shown in gray and blue) can effectively contain the true generalization errors of the model (for both $\epsilon = \{0.5, 1\}$). Furthermore, we can see that when the exploration probability $\epsilon$ is increased, the bounds get tighter (the blue line is below the gray line) due to the additional samples explored during data collection.

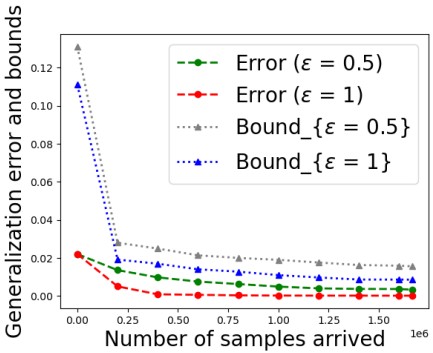

Figure 8: Generalization error and bounds using *Retiring Adult* dataset.

|  |  | Number of Samples Arrived | | | | | |
|---|---|---|---|---|---|---|---|
|  |  | 0 | 0.2M | 0.4M | 0.8M | 1.2M | 1.6M |
| $\epsilon = 1$ | $|R(\hat{\theta}) - R_{emp}(\hat{\theta})|$ | 0.0220 | 0.0050 | 0.0009 | 0.0003 | 0.002 | 0.0002 |
|  | Generalization Error Bounds | 0.111 | 0.019 | 0.017 | 0.013 | 0.010 | 0.009 |
| $\epsilon = 0.5$ | $|R(\hat{\theta}) - R_{emp}(\hat{\theta})|$ | 0.0220 | 0.0136 | 0.0098 | 0.0063 | 0.0040 | 0.0037 |
|  | Generalization Error Bounds | 0.131 | 0.028 | 0.025 | 0.020 | 0.0176 | 0.0159 |

Table 3: Numerical summary table for the generalization error and bounds across the number of arrived samples using *Retiring Adult* dataset.

**Experiments on *Adult* dataset.** The *Adult* census dataset is similar to the *Retiring Adult* dataset, but it has smaller amount of samples. It is also used to predict whether an individual can earn more than $50k/year, based on a multi-dimensional feature set. A total of 45000 new samples arrive throughout the experiment; We report our experiment results for an average of 5 runs, where the randomness comes from the order of samples arrived and the exploration. In addition, we further consider the model is updated as new samples are collected. Therefore, in the following experiments using *Adult* dataset, we also assess the performance of our bounds based on whether we adaptively update the decision threshold $\hat{\theta}$ with new samples.

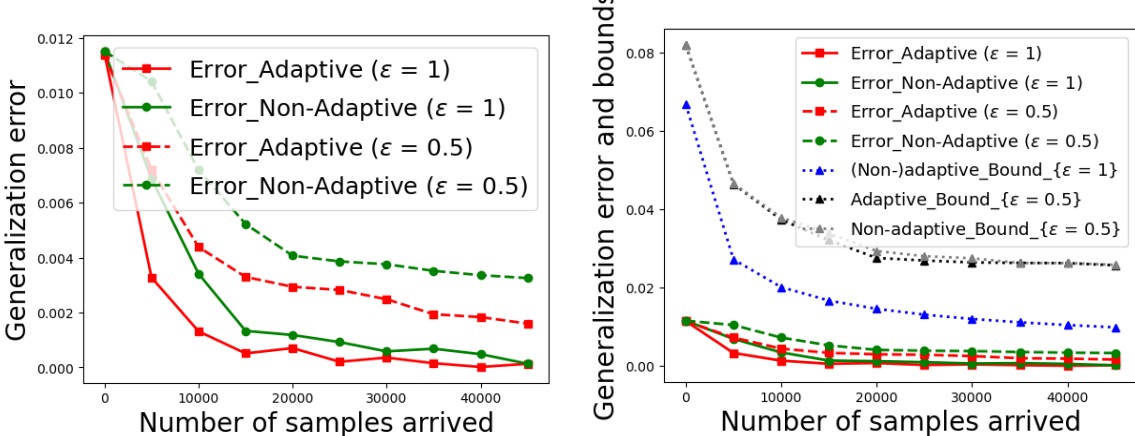

Figure 9: Generalization error with(out) an adaptively updated model $(\hat{\theta})$ and varying exploration $(\epsilon)$.

| | | Number of Samples Arrived | | | | | |
|---|---|---|---|---|---|---|---|
| | | 0 | 5k | 15k | 25k | 35k | 45k |
| $\epsilon = 1$ | $\|R(\hat{\theta}) - R_{emp}(\hat{\theta})\|$ | 0.0114 | 0.0069 | 0.0013 | 0.0009 | 0.0007 | 0.0001 |
| | $\|R(\hat{\theta}) - R_{emp}(\hat{\theta})\|$ (Adaptive) | 0.0114 | 0.0033 | 0.0005 | 0.0002 | 0.0002 | 0.0001 |
| | Generalization Error Bounds | 0.067 | 0.027 | 0.017 | 0.013 | 0.011 | 0.010 |
| $\epsilon = 0.5$ | $\|R(\hat{\theta}) - R_{emp}(\hat{\theta})\|$ | 0.0014 | 0.0104 | 0.0052 | 0.0039 | 0.0035 | 0.0033 |
| | $\|R(\hat{\theta}) - R_{emp}(\hat{\theta})\|$ (Adaptive) | 0.0114 | 0.0072 | 0.0033 | 0.0028 | 0.0019 | 0.0016 |
| | Generalization Error Bounds | 0.082 | 0.046 | 0.034 | 0.028 | 0.026 | 0.026 |
| | Generalization Error Bounds (Adaptive) | 0.082 | 0.046 | 0.032 | 0.027 | 0.026 | 0.026 |

Table 4: Numerical summary table for the generalization error and bounds across the number of arrived samples using *Adult* dataset.

From Figure 9, the y-axis represents the generalization error and its bounds, where a lower value is preferable. A smaller value of the bounds indicates tighter bounds enclosing the generalization error curve. We observe that as the decision threshold $\hat{\theta}$ is adaptively updated when more samples are collected, it has even better generalization performance compared to a non-adaptive decision threshold (evidenced by the red curve being lower than the green curve) This is expected as a refined decision threshold yields better performance on unseen data. Further, for the generalization error bounds (dotted lines in the right panel), we see that our bounds effectively contain the true generalization errors of the model for both the fixed model and adaptively updated model cases (all dotted lines are above the red/green curves). Notably, in the presence of censored feedback, we observe that the generalization error bound with adaptive updating is tighter than the non-adaptive one (the black curve is below the gray curve), pointing to a potential future research direction for further improving our bounds.

### 5.3 Comparison with existing generalization error bounds

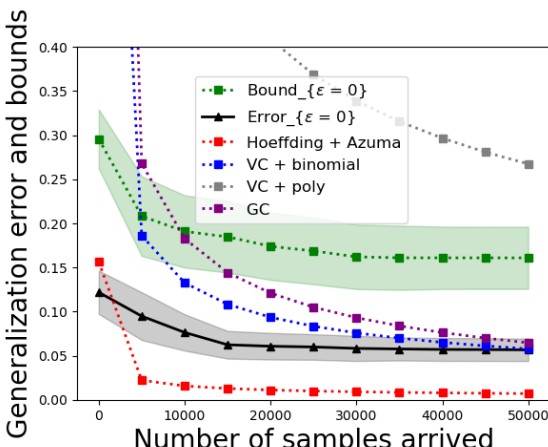

Figure 10: Existing bounds fail to capture generalization when there is censored feedback.

We now compare the performance of our bounds with a number of existing generalization error bounds, and show that by failing to account for censored feedback, prior works fail to correctly capture how well a model learned on data suffering from censored feedback generalizes to unseen data. We consider the following four benchmarks: The 'Hoeffding + Azuma' bounds represent those derived from Hoeffding and Azuma inequalities (Hoeffding, 1994; Azuma, 1967). The 'VC + binomial' bounds are VC generalization bounds (Vapnik & Chervonenkis, 2015; Abu-Mostafa et al., 2012, Thm 2.5) where the shatter coefficient is bounded through the binomial theorem. The 'VC + poly' bounds represent VC generalization bounds (Vapnik & Chervonenkis, 2015; Devroye et al., 2013, Thm 13.11) applicable to any linear classifier whose empirical

| Existing Generalization Bounds and Errors | Number of Samples Arrived | | | | | |
|---|---|---|---|---|---|---|
| | 0 | 10k | 20k | 30k | 40k | 50k |
| $\|R(\hat{\theta}) - R_{emp}(\hat{\theta})\|$ | 0.122 | 0.076 | 0.061 | 0.058 | 0.057 | 0.057 |
| Hoeffding + Azuma Bounds | 0.156 | 0.016 | 0.011 | 0.009 | 0.007 | 0.007 |
| VC + binomial Bounds | 1.128 | 0.562 | 0.409 | 0.339 | 0.296 | 0.267 |
| VC + poly Bounds | 4.453 | 0.133 | 0.093 | 0.076 | 0.065 | 0.058 |
| GC Bounds | 1.827 | 0.183 | 0.121 | 0.093 | 0.076 | 0.065 |
| Our Bounds | 0.296 | 0.191 | 0.174 | 0.162 | 0.161 | 0.161 |

Table 5: Numerical summary table for comparisons with existing bounds across the number of arrived samples.

error is minimal, where the shatter coefficient is bounded by a polynomial function. Lastly, the 'GC' bounds (Glivenko, 1933; Cantelli, 1933) are derived based on the Glivenko-Cantelli Theorem for a threshold classifier and 0-1 loss.

We conduct this experiment on synthetic data. We start with 50 initial training samples for each label $y \in \{0, 1\}$ randomly drawn from Gaussian distributions N(9,1) and N(10,1), respectively. The decision threshold $\hat{\theta}$ is selected to be the one minimizing the misclassification error on the training data. Then, a total of 50000 new samples arrive throughout the experiment. They will be accepted if the feature $x \geq \hat{\theta}$, otherwise, they are rejected. We run the experiments 5 times and report the average results with corresponding error bars. From Figure 10, we can clearly see that the 'Hoeffding-Azuma' (red), 'VC+binomial' (blue), and 'GC' (purple) bounds are inadequate for accurately estimating the true generalization error guarantees of the model. This inadequacy is demonstrated by the fact that all three bounds cross the true error (black) line as new samples are collected under the presence of censored feedback. For the 'VC+poly' (gray) bound, although it provides a very loose estimate compared to our bounds for the given number of new samples—evidenced by the gray bounds being above our green bounds—it ultimately exhibits similar behavior to the other three benchmarks, in that it will go lower than the true generalization error.

## 6 Conclusion and Future Work

We studied generalization error bounds for classification models learned from non-IID data collected under censored feedback. We presented two generalizations of the Dvoretzky-Kiefer-Wolfowitz (DKW) inequality, which characterizes the gap between empirical and theoretical CDFs given *IID* data, to problems with *non-IID* data due to censored feedback without exploration (Theorem 2) and with exploration (Theorem 3), and connected these bounds to generalization error guarantees of the learned model (Theorem 4). Our findings establish the extent to which a decision maker should be concerned about censored feedback's impact on the learned model's performance guarantees, and show that a minimum level of exploration is needed to alleviate it.

For future work, we are interested in strengthening our bounds by allowing the model ($\theta$) to be adaptively updated as new samples are collected; as noted in Section 5, this could help further strengthen our error bounds. Generalization error bounds under a combination of censored feedback and domain adaptation are also worth exploring, wherein the initial training data distribution differs from the target domain distribution. Finally, we have provided extensions of the DKW inequality, which strengthens the VC inequality when data is real-valued, under censored feedback; providing similar extensions of the VC inequality for *multi-dimensional data* could be an interesting direction of future work. We discuss some initial findings and potential challenges of this extension below.

**Bounds for higher dimensional data**. When assessing generalization error under censored feedback in higher dimensional data, one approach could be to first reduce the dimensionality, enabling direct application of our findings. For instance, we have performed a mapping of multi-dimensional features to a single-dimensional representation in our experiments on the real-world *Adult* census dataset. However, this reduction may lead to some loss of information, potentially impacting algorithm performance. An alternative

would be to follow our approach of identifying IID subspaces in the higher-dimensional data space, apply a *multivariate* DKW inequality (e.g., (Naaman, 2021)) in these subspaces, and then identify the appropriate error coefficients to re-assemble the subdomain bounds and find a CDF error bound for the entire data domain. We provide an analysis for 2D spaces based on this approach in Appendix J. A main challenge when doing so is that while the decision boundary can be any arbitrary line (determining the two subspaces in which data can be viewed as IID), the standard joint CDF calculates the probability that $X \leq x$ and $Y \leq y$, where $x$ and $y$ are vertical and horizontal cutoff values. To circumvent this mismatch, we start with an *adjusted* CDF which measures data density and counts existing vs. newly collected samples in a "rotated" data space, and subsequently map the CDF error bound of the adjusted CDF to a CDF error bound for the standard CDF (as detailed in Appendix J). Alternative error bounds that build on the VC inequality for multi-dimensional data (instead of multi-dimensional DKW inequalities), remain as a potential direction for future work.

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
