# OpenReview forum: "Generalization Error Bounds for Learning under Censored Feedback"
_TMLR — Rejected by TMLR_

### Review · Reviewer_2hkU · 2024-11-15

**Summary Of Contributions:**

The paper provides worst case bounds for the maximum distance between the empirical c.d.f. and the groundtruth c.d.f. of random variables under censored feedback (i.e. when data from a region of the covariate space is not observed). The main idea and bounding technique consists on bounding the error on the two regions and the combining the two error using a union bound. The authors also consider a "relaxed" censored formulation, in which data from the a portion of the censored region can be sampled with some probability.

**Audience:**

Yes

**Claims And Evidence:**

No

**Requested Changes:**

I believe the paper has potential to deliver interesting results; however, as it stands, the setting and assumptions are not clearly defined. The same applies to the correctness of the results, which, based on the current description of the setup, do not appear to hold. As such, I suggest the authors revise the manuscript to clarify the setup, explicitly state all assumptions, and provide a rigorous justification for the claimed results. Additionally, ensuring consistency in terminology and notation throughout the paper would greatly improve its readability.

**Strengths And Weaknesses:**

I find the setting and experimental results presented in the paper to be interesting and relevant. The section on exploration strategies is interesting

I strongly encourage the authors to provide a formal characterization of the algorithm and address the contradicting and imprecise statements. The primary concern revolves around the dependence of $\theta$ on the initial dataset. On one hand, multiple parts of the paper mention that $\theta$ is used to select a binary classifier and determine the censored regions. However, for Theorem 2 and Theorem 3 (the main results of the paper) to hold, it appears that \theta must be independent of the initial dataset. Otherwise, the distribution of m would not be binomial with parameter $\alpha$, as implicitly assumed.

Other points requiring clarification:
- “Denote the portion of qualified (resp. unqualified) samples in the population by $p_1$.” Is this referring to $\Pr[Y=1]$ or $n_1$?
- $F^y(x)$ is described as the feature distribution but appears to actually be the c.d.f. of the features.
- I recommend avoiding the terms “decision maker” for the learning algorithm and “agents” for the data points. Why not simply use “learner” and “data samples” instead?
- In Figure 5 and 6 the x-axis labels is "Number of samples arrived", isn't this the size of the training data set?

---

> ### Author Response · Authors · 2025-01-29
> **Response to Reviewer 2hKU**
>
> Thank you for your thorough review and thoughtful comments on our manuscript. We have carefully addressed your suggestions, revising the manuscript accordingly. The changes made are highlighted in blue in the revised draft, and we provide detailed responses below.
>
> **1. Notations of $p_1$:** Thank you for pointing this out. $p_1$ refers to $\mathbb{P}(Y = 1)$. We have revised the Problem Setting section to include clearer explanations and have added a notation summary table at the end of the section for better clarity.
>
> **2. Notations of $F^y(x)$:** We appreciate your attention to detail regarding this typo. It has been corrected in the revised manuscript.
>
> **3. Terminologies of decision maker and agent:** Thank you for this comment. We have revised the terminology to enhance clarity and maintain consistency. Following your suggestion, we now refer to the learning algorithm as the "learner" and use "(data) samples" to describe the data points.
>
> **4. X-axis information:** Thank you for your comment. The x-axis in Figures 9 and 10 (formerly Fig. 5 and 6) represents the number of new samples that have arrived. This can be interpreted as $T$, which may either be fixed in advance or denote the time at which a specific number of samples have been collected. Note that the size of the training dataset is smaller than the number of arriving samples, as some samples may not be explored.
>
> **Responses on requested changes:** We are grateful for your careful reading of our paper. Based on your suggestions and feedback from other reviewers, we have revised Section 2 (Problem Setting) extensively. Specifically, we now explicitly state our assumptions and include discussions/justifications for them. Additionally, we have added a notation summary table at the end of Section 2 to improve the clarity of our notations. We also revise our Theorems accordingly. We hope that these revisions, along with our responses, address your concerns and enhance the quality of our manuscript. We are happy to answer any further questions, or provide additional clarifications and revisions.

---

### Review · Reviewer_k1kD · 2025-01-11

**Summary Of Contributions:**

This paper proposes a new theoretical analysis of generalization error bounds with non-IID data caused by censored feedback.
The analysis, which is an extension of the Dvoretzky-Kiefer-Wolfowitz (DKW) inequality to non-IID data, reveals that traditional generalization error bounds fail to account for censored feedback, underscoring the necessity of the proposed bounds. Furthermore, the paper evaluates the impact of exploration techniques, such as pure and bounded exploration, on improving error bounds. These findings guide decision-makers in balancing the trade-off between enhancing generalization guarantees and the costs of data collection under censored feedback constraints.

**Audience:**

Yes

**Broader Impact Concerns:**

There is no concern about the ethical implications.

**Claims And Evidence:**

Yes

**Requested Changes:**

1. The topic is related to online learning with partial feedback (e.g., multi-arm bandit) problems. Please add the discussion in the related work.
2. Theorem 3 is complicated. Please add more annotations or explanations so that readers can more easily understand the meaning of each term.
3. Please add more descriptions to the figures. Such as how to interpret the results and the take-home message in each figure.
4. The meaning of m is not described in Theorem 2, although it is defined in the text before. In general, there are many notations, and the reader may not remember the meaning of each variable. I suggest that the authors put a table of notations and their meanings for easier reference.

**Strengths And Weaknesses:**

Strength:
- The problem is important and interesting.
- The problem is rigorously defined.
- The paper is well-written.
- Most of the theoretical analysis seems correct and solid.

Weakness
- The analysis only focuses on single-dimensional data. Although the authors claim that single-dimensional features and threshold classifier assumptions are not too restrictive, they still limit the usefulness of the analysis in practice.
- Some bounds (e.g., Theorem 3) are complicated and ugly, which makes them harder to be used to gain insights.
- Figures are hard to understand. For example, in Figure 6, what kind of curve means better? Higher, lower, or closer to the green curve?

---

> ### Author Response · Authors · 2025-01-29
> **Response to Reviewer k1kD**
>
> Thank you for your careful review and comments regarding our manuscript. We have revised the manuscript following the comments and highlighted the changes we made in blue in the new draft. We also detail these changes below.
>
> **Weakness 1: On one-dimensional Data.** Thank you for your detailed reading and valuable feedback. We agree that the analysis in the main body of the manuscript primarily focuses on single-dimensional data. However, we extend our analysis to higher-dimensional samples in Appendix J, using the multivariate DKW inequality. The key challenge in higher-dimensional settings (e.g., 2D space) lies in the fact that the decision boundary can be an arbitrary line, which determines the two subspaces in which data can be considered IID. In contrast, the standard joint CDF evaluates the probability $\mathbb{P}(X \leq x, Y \leq y)$, where $x$ and $y$ represent horizontal and vertical thresholds. As discussed in the conclusion section and Appendix J, we propose an adjusted CDF as a proxy to derive error bounds for the standard CDF in high-dimensional cases. To enhance the flow of our paper, we refer to Appendix J for high-dimensional samples when we made our one-dimensional sample assumption in Assumption 1.
>
> **1. Additional discussion on the MAB.** Thank you for your suggestion regarding additional literature on MAB. We agree that our topic is related to online learning with partial feedback (e.g., multi-arm bandit) problems. While we previously included this discussion in Appendix A (Additional and Detailed Related Work), following your suggestion, we have now moved and revised it to the main body of the paper to provide a more comprehensive literature review.
>
> Specifically, our work is related to the broader literature on multi-armed bandit learning [1,2], which deals primarily with the exploration and exploitation dilemma. This same trade-off emerges the context of online/machine learning, where a decision maker can obtain additional data to improve the generalization performance of its models, while at the same time risking incurring costs due to this data collection. In the general bandit problem, the decision maker explores "arms" (the available actions) in various ways, such as randomly using the $\epsilon$-greedy algorithm, by some form of highest uncertainty as in UCB algorithm, or by importance sampling approaches as in EXP3, etc. The key difference in our approach is that we consider \emph{bounded} exploration (motivated by works such as [3,4,5,6]), where a bound is set to limit the ``arms'' that are considered for exploration (here, ranges of data samples that may be admitted). This is because the cost of wrong decisions increases as samples further away from the current decision threshold are admitted, making some arms too costly for exploration. Furthermore, in the bandit literature, regret analysis is conducted to analyze the model performance compared to the best actions in hindsight. In contrast, we analyze the model performance from a different angle: our goal is to improve the generalization error guarantees (upper bound on the difference between the model's performance on training data and unseen testing data) by utilizing the newly collected samples through exploration.
>
> [1] Sébastien Bubeck, Nicolo Cesa-Bianchi, et al. Regret analysis of stochastic and nonstochastic multi-armed bandit problems. Foundations and Trends® in Machine Learning 5.1 (2012): 1-122.
>
> [2] Tor Lattimore and Csaba Szepesvári. Bandit algorithms. Cambridge University Press, 2020
>
> [3] Maria-Florina Balcan, Andrei Broder, and Tong Zhang. Margin based active learning. In Learning Theory: 20th Annual Conference on Learning Theory, COLT 2007, San Diego, CA, USA; June 13-15, 2007.
> Proceedings 20, pp. 35–50. Springer, 2007.
>
> [4] Cheolhei Lee, Kaiwen Wang, Jianguo Wu, Wenjun Cai, and Xiaowei Yue. Partitioned active learning for heterogeneous systems. Journal of Computing and Information Science in Engineering, 23(4):041009, 2023.
>
> [5] Dennis Wei. Decision-making under selective labels: Optimal finite-domain policies and beyond. In International Conference on Machine Learning, pp. 11035–11046. PMLR, 2021.
>
> [6] Yifan Yang, Yang Liu, and Parinaz Naghizadeh. Adaptive data debiasing through bounded exploration. Advances in Neural Information Processing Systems, 35:1516–1528, 2022.

---

> > ### Author Response · Authors · 2025-01-29
> > **(Cont.) Response to Reviewer k1kD**
> >
> > **2. Explanations on Theorem 3.** Thank you for your suggestion to enhance the readability of Theorem 3 by providing additional explanations. We have revised Theorem 3 in the main body of the paper, where $m$ (the number of samples that fall below the decision threshold) is treated as a constant. To address the randomness in $m$, as discussed in an earlier round of review, we have moved this analysis to Appendix K. Additionally, we have included further explanations at the end of Theorem 3 to clarify its implications.
> >
> > Specifically, Theorem 3 provides an extension of the DKW inequality to account for censored feedback and exploration, introducing three distinct regions: the (still) censored region $(-\infty, LB)$, the exploration region $(LB, \theta)$, and the disclosed region $(\theta, \infty)$. The (still) censored region contributes a constant error term dependent on $l$, the number of initial samples in this region, due to the absence of exploration. The exploration region introduces $k_e$, the number of samples collected under an exploration probability $\epsilon$, which reduces the error in this region as $k_e \rightarrow \infty$, ultimately approaching zero. In contrast, the disclosed region contributes an error based on $n-m$, the number of initial samples above $\theta$, and $k_d$, the number of additional samples collected in this region. As $k_d$ increases, the error in the disclosed region also diminishes.
> >
> > A key insight from Theorem 3 is that although there can still be a non-vanishing error term in the (still) censored region, additional samples collected in the exploration and disclosed regions can reduce their respective error terms. Similar to Theorem 2, due to the newly collected samples $k_e$ and $k_d$, the error term for the exploration and disclosed region decreases asymptotically, behaving as $2exp(-2k_e\eta^2)$ and $2exp(-2k_d\eta^2)$, respectively. However, as also noted in Fig. 5, the union bounds can be problematic when $k_e, k_d$ is small, where the red line initially is above the orange line with a small exploration probability. Further, if we adopt pure exploration ($LB\rightarrow -\infty$, which makes $\beta \approx 0, l \approx 0$), the first term will vanish as well (however, note that pure exploration may not be a feasible option if exploration is highly costly).
> >
> > **3. Figure description.** Thank you for your suggestion to provide more detailed descriptions of our results. We have added further discussions, clarified our findings, and revised our explanations as follows:
> >
> > In Fig. 1, we observe that the bounds will decrease to zero as more samples are collected across the entire data domain. However, in the presence of the censored feedback, the error term from the censored region cannot be improved. This results in a persistent gap between the theoretical and empirical CDFs, preventing convergence to zero. In contrast, the bounds derived from the DKW inequality, which assume random sampling across the entire data domain, continue to decrease to zero. As a result, the bounds represented by the black curve eventually underestimate the true generalization error, crossing below it and failing to account for the censored feedback.
> >
> > In Fig. 3, we observe that for any fixed initial training sample size $n$, the CDF bounds for the censored region remain constant. Since this term does not depend on newly collected samples, the bounds can only be improved by increasing $n$, which aligns with the findings in Fig. 1. On the other hand, for the disclosed region, the CDF bounds gradually vanish as more samples are collected, demonstrating the impact of additional data on reducing the gap in this region.
> >
> > In Fig. 5, we observe that the green line provides a tighter bound than the orange line, with both providing tighter bounds than the blue line. This is shown by that the green line is below the orange line, which is also below the blue line. This improvement is due to collecting more samples from the disclosed region results in a decrease in the CDF error bound, as noted by Proposition 1. Additionally, we can observe from the trajectory of the red line that introducing exploration enlarges the CDF error bound due to the additional union bound, but it also enables the collection of more samples, leading to a decrease in the CDF error bound as $\epsilon$ increases (evidenced by the red line is decreasing when $\epsilon$ increases along the x-axis); note that this observation aligns with Proposition 2.

---

> > > ### Author Response · Authors · 2025-01-29
> > > **(Cont.) Response to Reviewer k1kD**
> > >
> > > In Fig. 6,  we observe that our bounds effectively enclose the true distribution, evidenced by that the true distribution is upper and lower bounded by the dotted lines. We also note the distinction between empirical CDFs in the disclosed region ($x\geq7$) and the censored region ($x \leq 7$): as intuitively expected, empirical CDFs (solid lines) in the disclosed region are ``smoother'' compared to those in the censored region. Furthermore, as $\epsilon$ (exploration) increases, we overcome censored feedback in the exploration region, resulting in more accurate empirical estimates. Additionally, as $\epsilon$ increases, our error bounds improve (i.e., more tightly enclose the true CDF). In other words, we can see from Figure 6 that the empirical CDF in both explored and disclosed regions is getting smoother and closer to the true distribution with more samples collected. In addition, with a higher $\epsilon$, the gap between the upper and lower bounds is smaller (tighter) and can still enclose the true distribution.
> > >
> > > In Fig. 7 - 8, the y-axis represents the generalization error and its bounds, where a lower value is preferable, as it indicates tighter bounds enclosing the generalization error curve. We can observe that our bounds can effectively enclose the true generalization errors of the mode for both $\epsilon = \{0.5, 1\}$. Furthermore, we can see that when the exploration probability $\epsilon$ is increased, the bounds get tighter (the blue line is below the gray line) due to the additional samples explored during data collection.
> > >
> > > In Fig. 9 (formerly Fig. 5), we observe that as the decision threshold $\hat{\theta}$ is adaptively updated with the collection of more samples, it results in improved generalization performance compared to a non-adaptive decision threshold (evidenced by the red curve being lower than the green curve). This aligns with expectations, as a refined decision threshold typically performs better on unseen data. Additionally, for the generalization error bounds (dotted lines in the right panel), we observe that the bounds effectively enclose the true generalization errors of the model for both the fixed and adaptively updated cases (all dotted lines are above the red/green curves). Notably, in scenarios with censored feedback, the generalization error bound with adaptive updating is tighter than that of the non-adaptive (the black curve is below the gray curve). This highlights a potential future research direction to further improve these bounds.
> > >
> > > In Fig. 10 (formerly Fig. 6), we observe that the "Hoeffding-Azuma" (red), "VC+binomial" (blue), and "GC" (purple) bounds are inadequate for accurately estimating the true generalization error guarantees of the model. This inadequacy is demonstrated by the fact that all three bounds cross the true error (black) line as new samples are collected under the presence of censored feedback. For the "VC+poly" (gray) bound, although it provides a very loose estimate compared to our bounds for the given number of new samples—evidenced by the gray bounds being above our green bounds—it ultimately exhibits similar behavior to the other three benchmarks, in that it will go lower than the true generalization error.
> > >
> > > **4. Notations.** Thank you for your suggestion to include a table of notations. We have now added a table of notations summary at the end of Section 2 (Problem Setting). Specifically, we summarize all notations in Table 1 in the revised manuscript.

---

### Review · Reviewer_wJGC · 2025-01-15

**Summary Of Contributions:**

**summary**

This paper studies learning guarantees when training data is collected under censored feedback, i.e., only accepting some of the labels. The authors show that classical Dvoretzky-Kiefer-Wolfowitz (DKW) bounds—derived under i.i.d. data assumptions—are generally too optimistic in this selective labeling scenario. They extend DKW to non-i.i.d. censored data by partitioning the domain into i.i.d. dub-domains and then reassembling subdomain-specific bounds into a global bound. The resulting error terms reveal how increasing censorship degrades generalization guarantees, and how exploration (admitting a fraction of points below the threshold) can mitigate these effects.

**Audience:**

Yes

**Broader Impact Concerns:**

No concern

**Claims And Evidence:**

Yes

**Requested Changes:**

please see the weaknesses.

**Strengths And Weaknesses:**

**Strengths**

-- The paper is well-presented and is relatively straightforward to follow.

-- The paper presents a novel extension of DKW inequality to non-iid-ness due to censored feedback.

-- the results on exploration vs the cost trade-offs are insightful.

-- some (limited) numerical results are provided.

**Weaknesses**

-- While there is a short discussion of related literature (e.g., learning under covariate shift, time-series correlations, or partition-based active learning), the paper could benefit from more direct quantitative or conceptual comparisons to known bounds from these non-i.i.d. frameworks.

-- The new expressions in theorems 2, 3 and the related lemmas involve multiple layers of union bounds and sums over partitions. Especially for large Ns, these closed-form expressions might be unwieldy to directly compute. It would be helpful to provide approximate forms or asymptotic analyses to give more intuition on how the bounds behave for large sample sizes.

-- Given the complexity of the derived bounds, numerical illustrations could be super helpful. However, they are limited in your work.

--- Why the synthetic experiments are not consistent? different experiments use different sample sizes or thresholds.

--- For the real-world census dataset, what is x exactly, i.e. what feature is selected for filtering?

--- It would help to include a brief numeric summary table (e.g., the actual maximum error gap, or the proportion of times the bound successfully encloses the true CDF) alongside the figures for section 5. This would give readers a more concrete sense of how tight/loose the bounds are.

---

> ### Author Response · Authors · 2025-01-29
> **Response to Reviewer wJGC**
>
> Thank you for your careful review and comments regarding our manuscript. We have revised the manuscript following the comments and highlighted the changes we made in blue in the new draft. We also detail these changes below.
>
> **Weakness 1. Related works and conceptual comparisons.** Thank you for your suggestion regarding the related work. In the revised manuscript, we have expanded the discussion on the multi-armed bandit problem, as our work is closely related to online learning with partial feedback. Additionally, we have provided high-level conceptual comparisons to known bounds from these non-IID frameworks.
>
> Specifically, for the related literature on time-series correlations where the dependence between samples weakens over time. To address the vanishing dependence issue, these works consider building blocks within which the data can be viewed as IID. However, we differ in our reassembly method, in the source of data non-IIDness, and in our study of the impacts of exploration. Furthermore, our bounds also differ conceptually. While their bounds, based on the mixing parameter $\beta$, converge to zero by treating random samples across identified blocks as IID, our bounds are derived from threshold-based data collection. They are constructed by reassembling multiple IID blocks while explicitly accounting for the effects of censored feedback.
>
> For the related literature on partition-based active learning. Our work is similar to these studies in that we also consider (active) exploration techniques, and partition the data domain to build IID blocks. However, we differ in problem setup and analysis approach, and in accounting for the cost of exploration when we consider bounded exploration techniques. More specifically, their bounds are derived from the aggregation of multiple subdomains with requested labels (we refer to as exploration). The key distinction lies in data availability: while they can request labels from any subdomain without considering the cost of doing so, we are constrained to explore samples from certain subdomains due to the presence of censored feedback.
>
> Additionally, we have provided a more detailed and comprehensive literature review on generalization under non-IID data in Appendix A. However, to the best of our knowledge, the impact of non-IIDness caused by censored feedback has not been explored in prior work. We believe our study is the first to investigate generalization performance under censored feedback conditions, the need for which we motivate with a number of application area examples, and benchmarking against existing error bounds.

---

> > ### Author Response · Authors · 2025-01-29
> > **(Cont.) Response to Reviewer wJGC**
> >
> > **Weakness 2. Expressions and bound behavior.** Thank you for your suggestions regarding the expression and behavior of the bound. To enhance the clarity of our analysis on censored feedback, we have revised the paper to assume a constant $m$ (representing the number of samples in the censored region) in the main body of the paper. In such cases, the summation notation is removed from our bound expressions. However, following feedback from the earlier round of review, we also account for the randomness of $m$ in Appendix K. We motivate and contrast the fixed $m$ and the random $m$ assumptions in this appendix. We hope that this re-ordering of our findings has resulted in a cleaner form for the expression of our bounds in the main paper, as each term explicitly corresponds to contributions from specific regions in the sample space.
> >
> > Moreover, regarding the asymptotic analysis, for related Lemmas and Theorems, we observe that as $n$ (the number of initial samples across the entire data domain) becomes large, the maximum discrepancy between the theoretical and empirical CDFs decreases, following the strong law of large numbers. In such cases, the effect of censored feedback on the bounds becomes minimal, as the data distribution can already be well estimated. More interestingly, when the initial training dataset is small, censored feedback can have a substantial impact on the bounds. As the number of samples collected under censored feedback increases ($k \rightarrow \infty$), Theorem 2 shows that the error term associated with the censored region remains constant, as it does not depend on $k$. However, the error term for the disclosed region decreases asymptotically, behaving as $2exp(-2k\eta^2)$, similar to the DKW bound. Together, this results in an overall decreasing trend, which is also reflected in the numerical illustration in Fig. 5, where the orange line falls below the blue line. Similarly, in Theorem 3, the error term associated with the (still) censored region remains constant, as it does not depend on $k_e$ and $k_d$. However, due to the newly collected samples $k_e$ and $k_d$, the error term for the exploration and disclosed region decreases asymptotically, behaving as $2exp(-2k_e\eta^2)$ and $2exp(-2k_d\eta^2)$, respectively. However, as also noted in Fig. 5, the union bounds can be problematic when $k_e, k_d$ is small, where the red line initially is above the orange line with a small exploration probability. The results in Fig. 5 show that the exploration probability $\epsilon$ is around 10\% when the bounds with exploration outperform that without exploration, and it is around 20\% when the bounds with exploration perform closely to the bounds which observing all samples in the exploration range. In other words, as $k$ (or, $k_e, k_d$) becomes large, the censored region error remains constant, while other error terms will vanish to zero. Further, if we adopt pure exploration ($LB\rightarrow -\infty$, which makes $\beta \approx 0, l \approx 0$), the first term will vanish as well (however, note that pure exploration may not be a feasible option if exploration is highly costly).
> >
> > Regarding the behavior of the bound, insights can be drawn from the original DKW inequality. Specifically, the bounds decrease as more samples are collected across the entire data domain. However, due to censored feedback, the collected samples are no longer representative of the entire domain, as they are restricted to the disclosed region. Consequently, the bounds associated with the censored region remain unchanged, resulting in a persistent gap between the theoretical and empirical CDFs, which prevents convergence to zero. In contrast, the bounds derived from the DKW inequality, which assume random sampling over the entire data domain, continue to decrease to zero. As a result, these bounds eventually underestimate the true generalization error, crossing below it and failing to account for the censored feedback. This distinction highlights the impact of censored feedback on the bounds' behavior. We have updated the manuscript to include this additional discussion in Section 3 after the introduction of the DKW inequality.

---

> > > ### Author Response · Authors · 2025-01-29
> > > **(Cont.) Response to Reviewer wJGC**
> > >
> > > **Weakness 3. Numerical illustration.** Thank you for your comments on the limited numerical illustration. As partially addressed in our response to Weakness 2, we have added additional visualizations to enhance clarity. Specifically, Fig. 1 illustrates the behavior of the bounds derived using the DKW inequality, while Fig. 3 demonstrates the behavior of the more complex derived bounds. Additionally, we have conducted experiments on real-world datasets, including FICO and Retiring Adult, as presented in Figs. 7 and 8.
> > >
> > > Specifically, in Fig. 3, we observe that for any fixed initial training sample size $n$, the CDF bounds for the censored region remain constant. Since this term does not depend on newly collected samples, the bounds can only be improved by increasing $n$, which aligns with the findings in Fig. 1. On the other hand, for the disclosed region, the CDF bounds vanish as more samples are collected, demonstrating the impact of additional data on reducing the gap in this region.
> > >
> > > Regarding the additional experiments, the FICO dataset is used to predict whether an individual with a certain credit score will default on a loan. It includes one-dimensional features (credit scores) with a specific focus on the distribution of the credit scores in different populations. The Retiring Adult census dataset, similar to the Adult dataset, is used to predict whether an individual earns more than $50k/year, based on a multi-dimensional feature set. However, it has more data samples compared to the Adult dataset, and is based on more recent census datasets.
> > >
> > > **Weakness 4. Experiment consistency.** Thank you for your careful review. In our CDF error bounds experiment, we use a smaller sample size to better highlight the impact of exploration on the smoothness of the curve and its influence on the improvement of bounds. In contrast, for our benchmark comparison experiment, we utilize a larger sample size to evaluate the effectiveness of our proposed bounds and compare them with existing bounds. In other words, the error bounds on CDFs are best observed in smaller sample regimes, while the attainable generalization error bound is best assessed from bigger datasets. Other experiment settings, such as LB and exploration probability, remain consistent across all experiments in the paper.
> > >
> > > **Weakness 5. Census data information.** Thank you for your detailed reading. As partially addressed in Weakness 3, the goal of the Adult census dataset is to predict whether an individual earns more than 50k/year based on demographic information (e.g., age, education, race, etc.). For our analysis, we excluded non-informative features (e.g., fnlwgt, capital gain/loss) and mapped multi-dimensional features into a single-dimensional representation, denoted as $x$, to simplify the analysis.
> > >
> > > **Weakness 6. Summary table** Thank you for your suggestion to include a numeric summary table. We have added a summary table alongside the experimental results, including both the original experiments and the additional ones suggested earlier, to present the actual generalization error alongside our proposed bounds.

---

### Comment · Reviewer_2hkU · 2024-10-29
**Preliminary Feedback**

Dear authors, before continuing the review I would like to obtain a clarification about Theorem 2.

According to Stage 1, the learner receives $n$ data points $x_1,\dots,x_n$ sampled i.i.d. from the underlying distribution, let's define this set $\mathcal{D}_0$. Based on $\mathcal{D}_0$ the learner selects a cut-off threshold $\theta(\mathcal{D}_0)$ (note the dependency on $\mathcal{D}_0$). You then define $m$ as the number of data samples from from  $\mathcal{D}_0$ that fall below $\theta$ and $n-m$ those that fall above. Furthermore, you define $\alpha=\Pr[X\leq \theta]$, i.e. the probability that the input feature falls below $\theta$. So far nothing wrong, even though I strongly suggest as a general comment to make these definitions formal and precise as the current version is not.

Based on my understanding, there is a problem in Theorem 2. In fact, the proof relies on conditioning on the even $\Pr[m=i]$ for $i=0,\dots,n$. You calculate this probability assuming it is Binomial. However this is true if $\theta$ is independent on samples $x_1,\dots,x_n$  , i.e. independent on  the initial data set. What you actually need to compute is $\Pr[m=i|\theta(\mathcal{D}_0)]$ which is not distributed as a binomial of parameter $\alpha$. For example, define $\theta=\min_i{x_i}$ then $\Pr[m=0]=1$ and $\Pr[m>0]=0$, which is not clearly not binomial distributed with parameter $\alpha=\Pr[X\leq \min_i{x_i}]$.

Please correct me if I am wrong. If I am not, how does this reflects in the following Theorems?

---

> ### Author Response · Authors · 2024-10-31
> **Response on the preliminary feedback**
>
> Dear reviewer 2hkU,
>
> Thank you for your detailed reading. We appreciate your recommendation on the notation, and will revise our draft accordingly.
> Regarding the question: Indeed, as you have noted, in Theorem 2, our analysis of the CDF bound is based on a fixed decision threshold $\theta$. We have adopted this as it is also a common assumption adopted in the active learning literature (as detailed in Remark 1 in the draft). For instance, in [1] and [2], the initial decision thresholds are derived from a realized/known dataset. Intuitively, this is akin to considering an institution who has an initial (realized) training dataset at hand, which they are using to find a decision threshold; our methods are intended to help them assess the consequences of using this (realized) threshold for decision making going forward.
> In fact, in such cases, both $m$ (samples in the censored region) and $\theta$ (decision threshold) would be deemed to be constants. However, we have also considered the randomness in $m$ in the current work, moving beyond this common assumption in the literature, following recommendations from an earlier round of reviews.
>
> In particular, we have considered $\theta$ to be constant, but $m^y$ to be a (binomial) random variable. This is because $\theta$ is the optimal threshold found from a collection of training data points with *both* labels y=0 and y=1. There is randomness in the data with label 0 and 1, so that even if the threshold is fixed, $m^y$ (say, y=1) can vary, depending on what the data on the other label (say, y=0) is. In other words, the decision threshold in Theorem 2's framework is viewed as fixed and not dependent on the specific samples x_1, …, x_n, because we do not impose any assumptions on samples from the alternative label distribution; or, we assume the other label distribution can have followed any distribution, such that any given (known) $\theta$ might have eventually realized.
> We hope this helps clarify our problem setting and would be happy to elaborate more. Thank you.
>
> [1] Corinna Cortes, Giulia DeSalvo, Claudio Gentile, Mehryar Mohri, and Ningshan Zhang. Region-based active learning. In The 22nd International Conference on Artificial Intelligence and Statistics, pp. 2801–2809. PMLR, 2019.
>
> [2] Cheolhei Lee, Kaiwen Wang, Jianguo Wu, Wenjun Cai, and Xiaowei Yue. Partitioned active learning for heterogeneous systems. Journal of Computing and Information Science in Engineering, 23(4):041009, 2023.

---

> > ### Comment · Reviewer_2hkU · 2024-11-06
> >
> > Thanks for the quick reply! From your response, I understand that for the random variable $m$ to be binomially distributed and for Theorem 2 to hold (and, I assume, the following ones as well), the threshold $\theta$ must be constant and cannot depend on the initial data. Given this, what is the utility of the initial data set since it cannot be used to design $\theta$?

---

> > > ### Author Response · Authors · 2024-11-07
> > > **Response on the follow-up feedback**
> > >
> > > Thank you for your reply. Yes, indeed as you note, for the random variable $m$ to be binomially distributed in the theorems, $\theta$ has to be a constant. Regarding the initial dataset, having access to it allows a decision maker to (numerically) evaluate the consequences of using their training data to inform future decisions (as reflected in the generalization thresholds we have found in the theorems). That is, a decision maker finds the variables needed to evaluate the bounds in our theorems based on the initial dataset -- the dataset of each label $y$ determines its corresponding $n^y$, and the dataset on both labels determines a $\theta$ (which in turn determines the $\alpha$ in the binomial distribution of each label's $m^y$, and other terms containing $\alpha$).

---

> > > > ### Comment · Reviewer_2hkU · 2024-11-08
> > > >
> > > > Thanks! Based on the sentence "the dataset on both labels determines $\theta$ (which in turn determines the $\alpha$ in the binomial distribution of each label's $m^y$... " it looks like that $\theta$ is indeed a function of the initial dataset. If so, the my original concern still exists. More specifically, given an initial dataset $x_1,\dots,x_n$ and a data dependent threshold $\theta=f(X_1,\dots,X_n)$ with each $X_i\sim X$ and independent, how can you show that the probability $\Pr[X_i<f(X_1,\dots,X_n)]=\Pr[X<f(X_1,\dots,X_n)]$ for all $X_i$ in the initial data set? Conditioning on $\theta=f(X_1,\dots,X_n)$ changes the distribution of $X_i$ unless $\theta$ is independent on $X_1,\dots,X_n$.

---

> > > > > ### Author Response · Authors · 2024-11-15
> > > > > **Response on the follow-up feedback**
> > > > >
> > > > > Thank you for highlighting the confusion. In the revised draft, we will add further discussion to improve clarity and address the notation issues mentioned in earlier comments. As you pointed out, $\theta$ is the optimal decision threshold determined by minimizing the empirical risk (ERM) using the initial realized IID samples $x^y_1, …, x^y_n$ for both labels y in {0,1}. However, in our derivation of generalization performance in Section 4, we assume $\theta$ is independent of the random initial samples.
> > > > >
> > > > > **Why does the decision threshold $\theta$ is fixed?** As noted in prior responses, any given (known) $\theta$ might have eventually realized. For example, consider 1000 samples from label 0 and label 1, drawn from $N(7, 1)$ and $N(10, 1)$, respectively. The optimal decision threshold $\theta$ is calculated as 8.52. However, even if we fix the same realizations for label 0 samples, different realizations of label 1 samples will result in different thresholds. For instance:
> > > > >
> > > > > -  If 1000 label 1 samples are drawn from $N(11, 1)$, the optimal $\theta$ could be 9.03.
> > > > >
> > > > > -  If 1000 label 1 samples are drawn from $N(9, 1)$, the optimal $\theta$ could be 7.97.
> > > > >
> > > > > Therefore, without prior knowledge of the label 1 distribution, $\theta$ cannot be uniquely determined. However, imposing assumptions on label distributions could be overly restrictive. Consequently, we simplify the analysis by assuming $\theta$ is a fixed constant.
> > > > >
> > > > > **CDF error bounds**: Since $\theta$ is assumed to be fixed and depends on data from both label 0 and label 1 samples, we can treat $\theta$ as given and known for the purposes of deriving CDF error bounds. These bounds are independent of label-specific information. For instance, as shown in Fig. 4, after applying the DKW inequality, our derived CDF error bounds can successfully enclose the true data distribution.
> > > > >
> > > > > **Generalization error bounds**: Our benchmark comparison experiments also indicate that the generalization error bounds, derived from the CDF error bounds and accounting for censored feedback, can successfully enclose the true error.
> > > > >
> > > > > **Randomness**: We can always refer to the no randomness case as detailed in previous version. In that case, the initial dataset is used to compute all required terms, including $\theta$, $m$, and $n$. After these terms are established, the downstream analysis does not depend on the specifics of the initial dataset. However, following your comments that $\theta$ is dependent on initial label distribution (say y = 0), but not purely dependent on label 0 distribution, we conducted the following toy experiment: We consider 1000 fixed realizations of label 0 samples and 1000 random realizations of label 1 samples drawn from $N(7, 1)$ and $N(10, 1)$, respectively. With $\eta = 0.015$,
> > > > >
> > > > > -  For the dependent case (where $\theta$ and $m$ change with each realization of label 1 samples), the average and standard deviation of the bounds from 5 different realizations are (0.7748, 0.0167).
> > > > >
> > > > > -  For the independent case (where $\theta$ is fixed at 8.51, determined from one realization of label 1 samples and held as constant thereafter), the corresponding values are (0.8167, 0.0143).
> > > > >
> > > > > These results show that incorporating dependence can provide tighter bounds but requires knowledge of the data distributions for both labels. Therefore, for simplicity, we adopt the independence assumption in our analysis.

---

### Decision · Action_Editor_3ZDc · 2025-03-03

**Recommendation:** Reject

**Comment:**

First, let me apologize for the delayed decision on this paper. This paper has gone through two full reviewing rounds now, and while the reviewers note various positive elements of the paper, there remain significant issues with how the key technical results are presented. In this round, the reviewers are leaning towards acceptance, but in my opinion, the clarity issues are far too severe to merit acceptance. While the authors have clearly made some efforts to address the most severe issues present in the initial submission (and subsequent second submission), these are in my opinion insufficient. Let me give a few representative examples.

- Page 4: the paragraph called *"The learning algorithm"* is quite misleading and likely to cause the reader a great deal of confusion. TMLR is a machine learning journal, and when readers see statements like "learning algorithm" or "training dataset", they will assume that the contents are the learning problem of interest. In reality, the contents of this paragraph have absolutely no bearing on the main results, since $\\theta$ must be fixed in advance.

- Remark 1: based on critiques from various reviewers, the authors have added this remark to emphasize that everything is "fixed", but again, just a few paragraphs earlier, they are saying that the learning algorithm determines $\\theta$ based on a "training dataset", which any reader is going to interpret as meaning that $\\theta$ is random.

- Page 5: in the stage 1 description, the authors say that the learner gets initial data, and based on it selects a "fixed" threshold. Statistically, this is a nonsensical statement. Either the threshold is fixed (i.e., not set based on data), or it is based on data. You cannot have it both ways. Several reviewers have commented very critically on this point so far, and the authors still have not properly revised the paper. The authors clearly *want* the reader to imagine that $\\theta$ is set based on data, but in such a setting their results simply do not hold.

- Theorem 2: the authors have $n$ samples which are called *"initial data samples"*. Earlier in the stage 1 description, the data used to determine $\\theta$ were also called "initial data", which is obviously confusing since $\\theta$ cannot be depending on the $x\_{1},\\ldots,x\_{n}$ formulated in Theorem 2. Extremely confusing.

- Theorem 4: the authors say *"Let $\\theta$ be a fixed/realized initial decision threshold calculated through one realization of the initial training dataset such that $f\_{\\theta}(x) = \\mathbb{1}(x \\geq 0)$.* This statement again is really not acceptable in my opinion; either $\\theta$ is fixed, or it depends on data.

There are countless examples similar to the above ones, both in the paper and in the response to reviewers. Consider for example the authors' response to Reviewer 2hkU, in which they say the following:

> __CDF error bounds:__ Since $\\theta$ is assumed to be fixed and depends on data from both label 0 and label 1 samples, we can treat $\\theta$ as given and known for the purposes of deriving CDF error bounds.

The above statement is basically nonsensical in my opinion. It is "fixed" but it also "depends on data"? This is not appropriate exposition, and two full rounds of review have not cleared up the numerous issues with clarity, and clarity is critical for TMLR submissions. As such, I must recommend for this paper to be rejected.

**Audience:**

In principle, data is frequently processed over different stages in time, and IID assumptions most certainly break down in many real-world settings, and so error bounds for core empirical quantities like the usual empirical CDF are of natural interest. The topic thus is definitely of general interest, but I feel the valid results are not yet clear enough to be expected to be able to have an audience of meaningful size.

**Claims And Evidence:**

The main claims in this paper are centered around extending the DKW inequality (tail bound on the empirical CDF error for independent observations) to a specialized non-IID setting. This specialized setting is one in which the data is collected in two stages. In the first stage, a threshold is determined (in the paper, $\\theta$). In the second stage, additional data is collected, but a filter is applied to it (based on $\\theta$), which causes the full data set (stage 1 data plus filtered stage 2 data) to be non-IID. Despite this described setting, the main results are valid only when the threshold $\\theta$ is *fixed* in advance (i.e., not random). Having gone through two full rounds of review (the first resulting in a reject) and several interactions with reviewers, the authors appear to be aware of this; they say that everything in stage 1 is "fixed" in their revised Remark 1. That said, there are so many examples in the paper where they seem to want the reader to think that $\\theta$ can be selected based on random data, that the overall takeaways from the paper end up being quite opaque.